# *FedIndex*: Federated Domain Adaptation with Continuous Domain Indices

**Qingyang Yu**                                                                    *qyu13@stevens.edu*
*Department of Electrical and Computer Engineering*
*Stevens Institute of Technology*

**Hao Wang**                                                                       *hw488@cs.rutgers.edu*
*Department of Computer Science*
*Rutgers University*

**Qizhen Zhang**                                                                   *qz@cs.toronto.edu*
*Department of Computer Science*
*University of Toronto*

**Hao Wang**                                                                       *hwang9@stevens.edu*
*Department of Electrical and Computer Engineering*
*Stevens Institute of Technology*

**Reviewed on OpenReview:** *https://openreview.net/forum?id=fnbGFH0330*

## Abstract

Federated domain adaptation incorporates source clients' knowledge to improve the model performance on the target client under the coordination of the server, mitigating the impact of data insufficiency and domain shift. Existing federated domain adaptation (FDA) methods focus on domain adaptation with categorical domain indices (*e.g.*, "source" and "target"), while many real-world tasks involve domains with continuous domain indices. For instance, hospitals need to adapt disease analysis and prediction across patients via *age*, a continuous domain index in medical applications capturing the underlying relation between patient information and disease analysis. Prior FDA methods struggle with such tasks due to their ignorance of continuous domain indices. This paper proposes FedIndex to enable FDA with continuous domain indices. FedIndex performs adversarial domain adaptation across clients with the help of a global discriminator, aligning all domains' distributions. Our theoretical analysis demonstrates the capability of FedIndex to generate domain-invariant features across clients using continuous domain indices without accessing data on clients, simultaneously maintaining privacy preservation. Our empirical results show that FedIndex outperforms the state-of-the-art FDA methods on synthetic and real-world datasets.

## 1 Introduction

Federated domain adaptation (FDA) has been proposed as a compelling framework that mitigates the challenges of domain shifts and further resolves data insufficiency in federated learning (FL) (McMahan et al., 2017). FDA incorporates the concepts of domain adaptation, leveraging labeled data from clients with source domains (*i.e.*, source clients) and unlabeled data from clients with target domains (*i.e.*, target clients). For instance, issues such as limited data availability (Kamp et al., 2023) and domain shift caused by divergent data distributions in different regions (Mehrjou et al., 2022; Rajendran & Others, 2023), such as COVID-19 and its variants in Europe (GISAID, 2025), can be mitigated by FDA (Nananukul et al., 2024; Li et al., 2020b; Liu et al., 2021) by leveraging the data of large hospitals in densely populated areas (*i.e.*, source clients) to assist small hospitals in sparsely populated areas (*i.e.*, target clients). Thus, FDA has been widely used in real-world scenarios to resolve data insufficiency and domain shift.

A rich literature of FDA solutions has been proposed recently, exploring several methodologies, *e.g.*, adversarial domain adaptation (Peng et al., 2019b; Hong et al., 2021; Wu & Gong, 2021; Yang et al., 2022), and aligning data distributions indirectly (Yao et al., 2021; Feng et al., 2021; Liu et al., 2023; Peng et al., 2019b; Jiang et al., 2024). However, existing studies only focus on *categorical domain adaptation* where domain indices are just discrete labels such as "source" and "target," making them fail in real-world scenarios involving *continuous domain indices*, where domain indices play the role of a distance metric capturing the underlying relation between domains and the task. For instance, medical applications often require adapting diagnostic models to account for patients across different age groups, where *age* serves as a continuous domain index.[1] In such scenarios, existing FDA methods that only recognize categorical domain indices cannot generalize to support continuous domain indices and thereby fail to effectively capture fine-grained relationships, such as those between disease manifestation and patient information.

This paper proposes FedIndex, the first FDA framework that supports continuous domain indices, bridging the gap left by prior studies that overlooked such indices in domain adaptation across clients with continuously indexed domains in FL. *First*, to address the FL constraint that prohibits access to raw client data, including continuous domain indices, FedIndex adopts an adversarial learning framework wherein local encoder-predictor pairs extract data embeddings and upload them, along with domain indices, to the server for centralized training. This design enables FedIndex to utilize knowledge (*e.g.*, data embeddings and domain indices) from all domains without violating privacy constraints. *Second*, to support continuous domain adaptation in FL and handle both domain shift and client heterogeneity, FedIndex employs a global discriminator on the server that regresses continuous domain indices via a distance-based loss. By receiving data embeddings from all clients, this global discriminator forms an adversarial training framework against local encoders, enabling domain alignment in a way that resembles centralized domain adaptation, even though raw data remains decentralized. Furthermore, the use of continuous domain indices theoretically guarantees domain alignment at equilibrium. *Additionally*, to preserve privacy during training, particularly given the additional data transmission (*i.e.*, data embeddings and domain indices) compared to conventional FL methods (*e.g.*, FedAvg), FedIndex applies differential privacy (DP) to both data embeddings and domain indices. This DP mechanism protects FedIndex against attacks such as distribution inference attack (DIA) and membership inference attack (MIA), without compromising its domain alignment performance, supported by empirical results and theoretical analysis.

Our contributions are as follows:

- We propose FedIndex as the first work on continuous domain adaptation in FL to address domain shift and data insufficiency. We provide a theoretical analysis that FedIndex aligns continuously indexed domains of all clients at equilibrium and obtains equivalent domain alignment performance compared to the continuous domain adaptation methods trained in centralized settings.
- We enhance FedIndex's robustness with DP and prevent privacy leakage from attacks, such as MIA and DIA. Meanwhile, our theoretical analysis demonstrates that noise introduced by DP does not degrade FedIndex's domain alignment performance at equilibrium.
- We present empirical results on synthetic and real-world datasets, showing that FedIndex significantly improves performance over the state-of-the-art FDA methods on continuously indexed domains. In the RotatingMNIST dataset, FedIndex achieves a 35.1% improvement in accuracy compared to the best FDA baseline.

## 2 Background & Related Work

**Centralized Domain Adaptation (CDA)** focuses on adapting a model trained in one domain (the source domain) to another domain (the target domain) for specific tasks like prediction. The fundamental objective is to align the distributions of the source and target domains by either directly matching statistics (Tzeng et al., 2014) or utilizing an adversarial loss (Goodfellow et al., 2014; Tzeng et al., 2017; Kuroki et al., 2018; Zhao et al., 2017). However, these existing centralized domain adaptation (CDA) works typically focus on

---

[1]Representative scenarios for FDA with continuous domain indices include predicting the risk of brain stroke using patients' physiological data and age (Kelly-Hayes, 2010; Soriano, 2021), and predicting sleep apnea syndrome using breathing-signal data and patient ages (Quan et al., 1997; Zhang et al., 2018).

adaptation among categorical domains where the domain index is merely a label. When these methods are applied to continuously indexed domains, they fail to consider the underlying relation between domain indices and the task.

To bridge this gap, Continuously Indexed Domain Adaptation (CIDA) (Wang et al., 2020) leverages domain indices as a distance metric. This approach captures the similarity distance between domain indices and the task, thus extending traditional categorical domain adaptation techniques to continuously indexed domains.

**Federated Domain Adaptation (FDA)** seeks to address domain shift across clients and data insufficiency. In a FDA setting, $N$ source clients $\{C_i\}_{i \in [N]}$ have their source domains $\{\mathcal{D}_i\}_{i \in [N]}$, and one target client $C_T$ has target domains $\mathcal{D}_T$ where $T = N + 1$. The input data and label are denoted as $x$ and $y$, respectively. For the source domains $\{\mathcal{D}_i\}_{i \in [N]}$, each domain contains the labeled data $\{(x_j, y_j)_{C_i}\}_{j=1}^{n_i}$, where $n_i$ is the number of data samples in domain $\mathcal{D}_i$ and client $C_i$ has full access to its domains $\mathcal{D}_i$. For the target client, target domains are unlabeled, containing data $\{(x_j)_{C_T}\}_{j=1}^{n_T}$, where $n_T$ is the number of data in target domains $\mathcal{D}_T$. This adaptation transfers source clients' knowledge to the target client, enabling accurate label predictions $\{(y_j)_{C_T}\}_{j=1}^{n_T}$ for the target client's data while simultaneously preserving privacy.

Prior work has extended CDA to FL environments (Peng et al., 2019b), primarily aligning the distributions of source and target domains in FL scenarios. To achieve this, existing studies adopt various approaches, including indirect domain alignment methods and adversarial domain adaptation. Specifically, some studies simulate domain distributions and align them, such as Gaussian mixture model (GMM) (Yao et al., 2021). KD3A (Feng et al., 2021) minimizes domain discrepancy via knowledge distillation. UFDA (Liu et al., 2023) generates pseudo-data and aligns domains with the help of a voting mechanism. Prior work also utilizes gradient updates to align domains (Jiang et al., 2024). Beyond methods that align domains indirectly, adversarial domain adaptation is employed due to its strong theoretical insights (Zhang et al., 2019). FADA (Peng et al., 2019b) deploys a global discriminator and local domain identifiers to construct an adversarial learning framework. FADE (Hong et al., 2021) naively employs adversarial CDA methods on each client with momentum aggregation on the server. Additionally, some studies leverage an adversarial loss to facilitate domain adaptation (Wu & Gong, 2021; Yang et al., 2022). However, previous studies neglect the presence of continuous domain indices in real-world scenarios, and prior adversarial-learning-based methods overlook the privacy leakage risks associated with accessing data embeddings in FL. In contrast, our work advances FDA by incorporating the concepts of continuous indices, extending it to continuous domain adaptation and maintaining data privacy preservation simultaneously.

## 3 Preliminary

To extend categorical FDA to continuous FDA, we assume a set of continuous domain indices $\mathcal{U} = \{\mathcal{U}_i\}_{i \in [N]} \cup \mathcal{U}_T$ for the source domains and the target domain, and $\mathcal{U}$ is a part of metric space (*i.e.*, a metric such as Euclidean distance). Domain indices are denoted as $u$. Then, for the source domains $\{\mathcal{D}_i\}_{i \in [N]}$ and the target domain $\mathcal{D}_T$, the data takes the forms of $\{(x_j, y_j, u_j)_{C_i}\}_{j=1}^{n_i}$ and $\{(x_j, u_j)_{C_T}\}_{j=1}^{n_T}$ respectively.

A naive solution to address this problem is a naive federated adaptation of CIDA, FedAvg+CIDA (*i.e.*, simply apply CIDA on each client). However, such a naive solution fails to achieve optimal domain alignment because the global model of the naive solution is merely an aggregation of those trained on the limited local data of each client, unlike centralized CIDA, where the model is trained on the entire dataset. We theoretically prove such a statement in Theorem 5.1 and empirically show it in Fig. 1. We also include more relevant experimental results in Section 6.

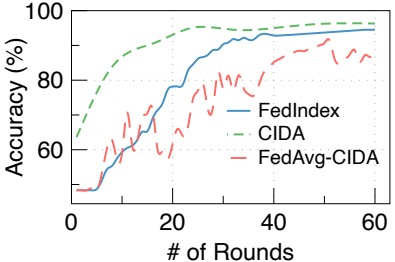

Figure 1: Preliminary accuracy results (%) on *Half-Circle* (Wang et al., 2020), a simple synthetic dataset used for binary classification tasks.

## 4 FedIndex's Design

### 4.1 Architecture

In contrast to the naive CIDA federation, to enable continuous domain adaptation across clients in FL with optimal domain alignment, we employ a global discriminator on the server predicting domain indices. Specifically, we employ one global discriminator $D$ alongside individual local encoder and predictor pairs $(E_i, F_i)$ for each client $C_i$. Furthermore, incorporating DP in FedIndex ensures data privacy and does not affect domain alignment at equilibrium.

In FedIndex, we aim to learn a global encoder $E_g$ and predictor $F_g$ such that the encoder $E_g$ aligns embedding distributions from all domains. Consequently, the global predictor $F_g$ can accurately predict data labels in target domains. Formally, given any embedding $z = E_g(x, u) \in \mathcal{Z}$, the global encoder $E_g$ can extract domain-invariant features such that $p(z|u_1) = p(z|u_2), \forall u_1, u_2 \in \mathcal{U}$. This objective is accomplished by training and aggregating local models from clients, with the global discriminator on the server facilitating the process. The min-max optimization for $(E_i, F_i, D)$ is:

$$\min_{E_i, F_i} \max_D V_p(E_i, F_i) - \lambda_d V_d(D, E_i), \tag{1}$$

where we have

$$V_p(E_i, F_i) \triangleq \mathbb{E}^s[L_p(F_i(E_i(x, u)), y)],$$

$$V_d(D, E_i) \triangleq \mathbb{E}[L_d(D(E_i(x, u) + \tau_i), u + \tau_u)],$$

where $L_p$ is the prediction loss taken over source clients (*e.g.*, cross-entropy loss for classification tasks) and $L_d$ is the domain index loss taken over all clients. $\lambda_d$ is the hyperparameter to balance two different loss functions. $\tau_i$ and $\tau_u$ are noises drawn from the Laplace distribution with mean $\mu = 0$ and scale $b = \frac{\Delta f}{\epsilon}$, where $\Delta f$ is the dataset sensitivity (commonly set to 1; see Appendix D.2 for detailed discussions) and $\epsilon$ is the privacy budget.

Eq. 1 defines the core adversarial objective of FedIndex, balancing two competing goals: preserving task-relevant information for prediction while removing domain-specific information from the learned representation. The first term, $V_p(E_i, F_i)$, is the prediction loss on labeled source data, which encourages the local encoder $E_i$ and predictor $F_i$ to retain features that are informative for the downstream task. The second term, $V_d(D, E_i)$, is the discriminator loss, where the global discriminator $D$ attempts to recover the domain index $u$ from the embedding produced by $E_i$. Since Eq. 1 is formulated as a min-max game, the discriminator is trained to minimize its domain index prediction error, while the encoder is trained adversarially to make such a prediction difficult. Consequently, the encoder is encouraged to produce embeddings that remain predictive of labels but are uninformative of the domain index, thereby aligning the feature distributions across domains. Thus, unlike conventional adversarial domain adaptation (DA) methods that only confuse a binary or multi-class domain classifier, Eq. 1 encourages the encoder to obscure the recoverability of a continuous domain coordinate, allowing FedIndex to handle evolving domain shifts smoothly.

The intuition behind introducing the global discriminator in Eq. 1 is that domain alignment in federated settings must be driven by information aggregated from all clients, rather than by isolated local alignment signals. A naive federated adaptation of continuous DA would only align domains locally on each client, and the subsequent model aggregation would generally fail to recover a globally aligned representation (see Section 5.1 for detailed demonstrations). By contrast, Eq. 1 enables the server-side discriminator to learn from embeddings uploaded by all clients and to provide a shared adversarial signal, which guides local encoders toward a representation space that is globally aligned across continuously indexed domains.

Besides, there are various designs for both the discriminator $D$ and $L_d$. For instance, $D$ can directly predict the domain index or the mean and variance of the domain index as in Probabilistic CIDA (Wang et al., 2020). In FedIndex, the discriminator $D$ directly predicts the domain index, and $L_d$ is $L_2$ loss for all clients $\{C_i\}_{i \in [T]}$:

$$L_d(D(z + \tau_i), u + \tau_u) = (D(z + \tau_i) - (u + \tau_u))^2. \tag{2}$$

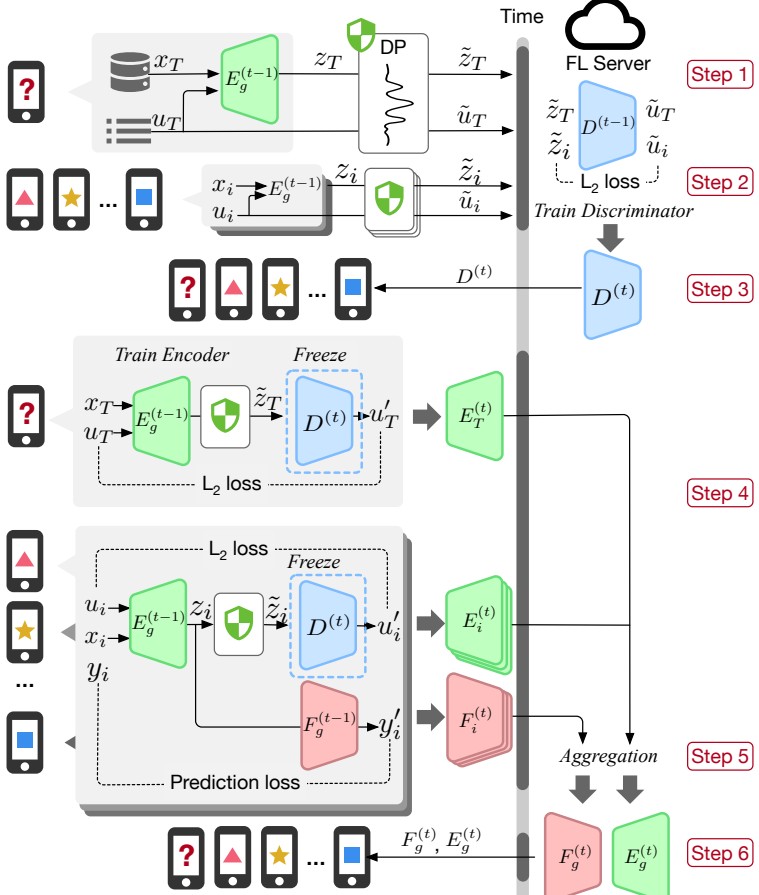

Figure 2: FedIndex's workflow in Round $t$.

## 4.2 Algorithm

FedIndex first initializes the local encoders $\{E_i\}_{i \in [T]}$ as $E_g^{(0)}$, local predictors $\{F_i\}_{i \in [N]}$ as $F_g^{(0)}$, and the global discriminator $D$ as $D^{(0)}$ at the server side. Then, FedIndex performs FDA with continuous domain indices $\mathcal{U}_i$ following Steps 1–6 in Figure 2 iteratively:[2]

**Step 1:** Each client $C_i$ (including target client) first adds Laplace noise $Lap(b)$ to the data embedding $z_i$ output by the local encoder $E_i^{(t-1)}$ and the domain indices $u_i$ (details in Section 4.3). Then, each client uploads noised data embeddings $\tilde{z}_i = z_i + \tau_i$ with noised domain indices $\tilde{u}_i = u_i + \tau_u$ to server $S$.

**Step 2:** The server $S$ trains $D^{(t-1)}$ with uploaded data and achieves $D^{(t)}$ by the max function of Eq. 1.

**Step 3:** The server $S$ sends $D^{(t)}$ back to all clients.

**Step 4:** Each client $C_i$ trains local encoder $E_i^{(t-1)}$ and predictor $F_i^{(t-1)}$ using predicted domain indices $u_i'$ and predicted labels $y_i'$ using the min function of Eq. 1. Then, each client $C_i$ receives $E_i^{(t)}$ and $F_i^{(t)}$ (target client $C_T$ only trains encoder $E_T^{(t-1)}$ using domain index loss since data in target domains is unlabeled). Then, each client $C_i$ uploads $E_i^{(t)}$ and $F_i^{(t)}$ to the server $S$ (target client $C_T$ only uploads $E_T^{(t)}$).

---

[2]Algorithm 1, provided in Appendix A, presents the complete workflow in pseudo-code.

**Step 5:** The server $S$ aggregates local models as global models $E_g^{(t)}$ and $F_g^{(t)}$ via the following rules:[3]

$$E_g^{(t)} = \frac{1}{T}\sum_{i=1}^{T} E_i^{(t)}, \ F_g^{(t)} = \frac{1}{N}\sum_{i=1}^{N} F_i^{(t)}. \tag{3}$$

**Step 6:** The server $S$ sends $E_g^{(t)}$ and $F_g^{(t)}$ to all clients for training in the next round as $E_i^{(t)}$ and $F_i^{(t)}$.

### 4.3 Privacy Preservation

Privacy preservation is a common consideration in studies relevant to FL and has been extensively explored by previous studies (Cao et al., 2022). However, FedIndex differs from conventional FL, such as FedAvg, by incorporating a global discriminator on the server, which processes data embeddings and domain indices for training.[4] This approach may expose sensitive information through data embeddings and domain indices. We focus on the data leakage issues introduced by such additional data transmission in FedIndex, where attackers (curious servers and clients) may attempt attacks such as DIA (Shokri et al., 2017b) and MIA (Shokri et al., 2017a) to obtain other clients' raw data. We employ DP (Dwork et al., 2006a) to defend such inference attacks by adding varying Laplace noises to transmitted data embeddings and fixed Laplace noise to transmitted domain indices in each communication round.[5] This distinction arises from the static nature of domain indices, which remain constant, in contrast to data embeddings that vary dynamically with changes in the encoders across communication rounds.

Other potential privacy concerns, such as gradient inversion, which have been extensively studied in prior research (*e.g.*, FLTrust (Cao et al., 2022)), are beyond the scope of this work and are not discussed here. Despite this, safeguards against other potential privacy issues are orthogonal to FedIndex and can be easily incorporated with FedIndex.[6] Theoretical analysis of DP in FedIndex is provided in Appendix C.1.

## 5 Theoretical Analysis

Directly applying CIDA to each client in FL scenarios leads to degraded domain alignment performance compared to centralized training, as demonstrated by both theoretical analysis and empirical results. To enable FedIndex to achieve domain alignment performance comparable to centralized CIDA, we build on the analytic framework introduced in prior work on adversarial domain adaptation (Zhao et al., 2019). Specifically, we adapt this framework to the FL setting to characterize the equilibrium behavior of the encoder, predictor, and discriminator, thereby revealing how FedIndex aligns domains. Our analysis shows that, at equilibrium, *FedIndex minimizes the impact of Laplace noise introduced by DP and achieves domain alignment performance comparable to that of centralized CIDA.* This analysis is presented under two scenarios:[7]

- **Simplified FedIndex**: This scenario involves only two types of components, the encoders $E_i$ and the discriminator $D$. The analysis focuses on the alignment of the discriminator $D$, the encoders $E_i$, and the global encoder $E_g$ at equilibrium.
- **FedIndex**: In practical applications, the ultimate goal is not just to align features but also to ensure that the model performs well on the target domain's task (*e.g.*, classification). Consequently, this analysis considers the interactions among the encoder $E_i$, the predictor $F_i$, and the discriminator $D$.

### 5.1 Domain Alignment Ability Analysis of the Naive Solution

This subsection presents Theorem 5.1, which formally demonstrates that the naive solution (simply employing CIDA on each client in FL scenarios) results in a system with degraded domain-alignment ability compared to CIDA trained in centralized settings. *Complete proofs are available in Appendix C.2.*

---

[3]For simplicity, we assign equal weights to all clients. In real-world settings, this can be readily extended by weighting clients proportionally to their number of local data samples relative to the number of global data samples.

[4]Appendix B.1 demonstrates the feasibility of uploading data embeddings to the server for further training.

[5]Algorithms 1 and 2 in Appendix A outline the security method in pseudo-code.

[6]Appendix B.2 provides the details of the incorporation of FedIndex and safeguards other than DP.

[7]The convergence analysis of FedIndex is in Appendix C.5

**Theorem 5.1** (Sub-Optimal Performance of the Naive Solution). *Naively applying CIDA to each client in FL scenarios results in a system with diminished domain alignment. Due to non-convexity, only with the help of locally trained discriminators do encoders align the conditional expectation of the domain index with the local marginal expectation within their respective domains, but fail to achieve global mean-alignment across all domains (i.e., $u \not\perp_1 z$).*

## 5.2 Analysis of Simplified FedIndex

The theoretical analysis follows the standard adversarial domain adaptation pattern, modeling a game in which the encoder aims to fool the discriminator, thereby preventing the discriminator from predicting the raw domain index. The theoretical analysis of the simplified FedIndex, in which each local client consists of an encoder $E_i$ without a predictor $F_i$, shows that FedIndex achieves optimal domain alignment compared to the naive solution. *Full analysis is available in Appendix C.3.*

In the absence of $F_i$, the optimization for $(E_i, D)$ is:

$$\max_{E_i} \min_D V_d(D, E_i) \tag{4}$$
$$= \mathbb{E}[L_d(D(E_i(x, u) + \tau_i)), u + \tau_u)].$$

where $\tau_i$ and $\tau_u$ are noises drawn from the Laplace distribution with mean $\mu = 0$. With a fixed encoder $E$ for all clients, the discriminator receives the noisy embedding $\tilde{z} = E(x, u) + \tau_i$. The optimal discriminator $D^*$ outputs the conditional expectation of the domain index given the noisy embedding:

$$D^*(\tilde{z}) = \mathbb{E}_{u \sim p(u|\tilde{z})}[u].$$

Assuming that $D$ always achieves the optimum w.r.t $E$, we can reformulate Eq. 4 as maximizing the discriminator's error, denoted as $C_d(E) \triangleq \min_D V_d(E, D)$. We then derive Theorem 5.2, demonstrating the global optimum of simplified FedIndex.

**Theorem 5.2** (Global Optimum for Simplified FedIndex). *With DP in FedIndex, $C_d(E)$ achieves the global optimum if and only if (1) the mean of Laplace noise terms are 0, i.e., $\mu(\tau_i) = \mu(\tau_u) = 0, \forall i$, and (2) the encoder $E$ satisfies the expectation of the domain index $u$ over the conditional distribution $p(u|z)$ for any given $z$ is identical to the expectation over the marginal distribution $p(u)$, i.e., $\mathbb{E}[u|\tilde{z}] = \mathbb{E}[u], \forall z$.*

**FedIndex versus CIDA:** One key difference between FedIndex and CIDA lies in the additive Laplace noise introduced by FedIndex's DP on the data embedding and domain indices. Our Theorem 5.2 proves that despite the additive Laplace noise, the global encoder still aligns the domain indices from all domains, thereby mitigating *domain shift*. Another key distinction between FedIndex and CIDA lies in their training dynamics (FL settings versus centralized settings). The aggregation process in FL renders conventional CIDA analysis patterns inapplicable, precluding a naive federation of CIDA from achieving optimal domain alignment (Theorem 5.1). To address this, we modify the system architecture by introducing a global discriminator on the server, enabling FedIndex to attain optimal domain alignment. Further details on this design are provided in Appendix B.4.

**Corollary 5.1.** *For FedIndex, the global optimum of $C_d(E)$ is achieved if the embeddings of all domains are mean-aligned, denoted as $u \perp_1 \tilde{z}$.*

**Corollary 5.2.** *For FedIndex, the Laplace noise introduced by DP does not shift the optimal alignment condition of the encoder, provided the noise has a mean of 0. The noise introduces a constant variance term $\mathbb{V}[\tau_u]$ to the objective function, which does not influence the gradients w.r.t the encoder.*

**Remark 5.1.** *For high-order alignment beyond the first moment (mean), it can be achieved by changing the loss function (e.g., from MSE to Gaussian NLL).*

## 5.3 Analysis of FedIndex

This subsection provides FedIndex's equilibrium states in the three-player game of $E_i, F_i$, and $D$ as defined in Eq. 1. We analyze the equilibrium behavior based on the dependency between the domain index $u$ and the label $y$.

$u \perp y$ (The Domain Index is a Nuisance Variable): The independence between the domain index $u$ and the label $y$ indicates that the domain index $u$ captures the nuisance variance instead of helpful information for prediction. Therefore, in this case, we prove the optimal global encoder captures all the information in the input $x$ relevant to the prediction while aligning the first moments (means) of the domain index distributions.

**Lemma 5.1** (Optimal Predictor)**.** *Given the encoder $E_i$, the prediction loss is lower-bounded by the conditional entropy:*

$$V_p(F_i, E_i) \triangleq L_p(F_i(E_i(x, u)), y) \geq H(y|E_i(x, u)),$$

*where $H(\cdot)$ is the entropy. The optimal predictor $F_i^*$ that minimizes the prediction loss outputs the true posterior probability:*

$$F_i^*(E_i(x, u)) = P_y(\cdot|E_i(x, u)).$$

Assuming the predictor $F_i$ and the discriminator $D$ achieve their optimum by Lemma 5.1, the local optimization (Eq. 1) can be rewritten as:

$$\min_{E_i} C(E_i) \triangleq H(y|E_i(x, u)) - \lambda_d C_d(E_i). \tag{5}$$

**Theorem 5.3.** *If the encoder $E_i$, the predictor $F_i$ and the discriminator $D$ are trained to reach optimum, any optimal local encoder $E_i^*$ has the following properties:*

$$H(y|E_i^*(x, u)) = H(y|x, u) \tag{6}$$

$$C_d(E_i^*) = \max_{E_i'} C_d(E_i') \tag{7}$$

Theorem 5.3 establishes that the optimal encoder retains all the information related to the label $y$ in the data $x$ and the domain index $u$ while simultaneously achieving alignment across different domains with noise data embedding and domain indices at equilibrium.

$u \not\perp y$ (The Domain Index is Informative): The domain index $u$ captures the information that helps predict label $y$. In this scenario, two terms in the objective function $C(E_i)$ conflict: (1) Minimizing $H(y|z)$ requires retaining information about $u$; (2) Maximizing $C_d(E_i)$ requires removing information about $u$. Consequently, FedIndex does not achieve perfect mean-alignment in this case. Instead, the hyperparameter $\lambda_d$ regulates a trade-off: the model learns a representation that is as domain-invariant as possible without sacrificing the predictive power derived from $u$. This prevents the model from relying on spurious domain correlations while retaining robust predictive features.

While Theorem 5.2 and Theorem 5.3 characterize the encoder, predictor and discriminator at their optimum, due to the non-convex nature of the optimization, this optimum might never be reached. Empirically, Section 6 shows that on standard datasets, FedIndex achieves improvement on model performance compared to baselines.

## 6 Evaluation

In this section, we evaluate FedIndex on two toy datasets (*Half-Circle* and *Sine*), an image dataset (*RotatingMNIST* (Wang et al., 2020)), and four real-world datasets (*CompCar* (Yang et al., 2015), *Brain-Stroke* (Soriano, 2021), *TCGA-BRCA* (The Cancer Genome Atlas Network, 2012), and *TPT-48* (Xu et al., 2022)). Meanwhile, we provide more comprehensive experimental results, including performance on standard DA benchmarks (*e.g.*, Office-Home (Venkateswara et al., 2017) and DomainNet (Peng et al., 2019a)) Furthermore, Appendix D provides visualization of training efficiency curves for each dataset, Kolmogorov-Smirnov (KS) tests for DP effectiveness, communication overhead analysis, and sensitivity tests regarding $\lambda$, $\epsilon$, and *epoch*. These empirical studies verify our theoretical analysis in Section 5 and show that:

- Previous categorical FDA methods result in sub-optimal alignment on continuously indexed domains. FedIndex can achieve an accuracy 13.37% higher on *Half-Circle* and 35.1% higher on *RotatingMNIST* than SOTA solutions (see Sections 6.2-6.8).

- The naive federated adaptation of CIDA (FedAvg+CIDA) achieves suboptimal domain alignment compared to FedIndex (see Table 1 and Table 2).
- FedIndex effectively aligns the continuously indexed domains in a manner comparable to CIDA trained in centralized settings, even with Laplace noise introduced by DP, demonstrating superior performance in different non-IID settings (see Table 1 and Table 2).
- FedIndex would not significantly increase communication overhead compared to conventional FL methods, such as FedAvg (see Table 10).
- FedIndex can ensure robust data embeddings and domain indices security via DP (see Appendix D.2).

## 6.1 Experiments Setup

**Datasets:** *Half-Circle* dataset comprises 30 domains, indexed from 1 to 30, arranged in a half-circle. Each domain contains 100 data samples $(x, y, u)$, where $x = (x_1, x_2)$ represents position coordinates (*e.g.*, $(x_1, x_2) = (0.1, 0.3)$), $y$ is a binary label (0 or 1), and $u$ is the domain index ranging from 1 to 30. *Sine* dataset, which follows a sine function-like data distribution, consists of 12 domains, each containing 190 data samples in the same format as the *Half-Circle* dataset $(x, y, u)$. *RotatingMNIST* is generated by continuously rotating MNIST from 0° to 180°. *CompCar* dataset is a real-world dataset of car images, annotated with attributes such as car types, viewpoints, and years of manufacture (YOM). *BrainStroke* dataset is another real-world dataset, featuring attributes relevant to brain stroke prediction. *TCGA-BRCA* dataset is a real-world dataset, featuring attributes relevant to breast cancer and survival prediction. *TPT-48* is a real-world dataset comprising monthly average temperatures for 48 states from 2008 to 2019. Please refer to Figure 3 in Appendix D for data visualization of *Half-Circle* and *Sine*.

**Baselines and Implementation:** We involve the following CDA and FDA models as baselines: CIDA (Wang et al., 2020) serves as the baseline in centralized machine learning (ML) and performs domain adaptation from source domains to target domains. MDD (Zhang et al., 2019) merges data into one source and target domain. DANN (Ganin et al., 2016) divides the domain spectrum into several domains and performs adaptation between multiple source and target domains. FedDA (Jiang et al., 2024) updates the global model's gradient via the convex combination of model gradients from the target client and source clients. FedGP (Jiang et al., 2024) is similar to FedDA, but model gradients of source clients are projected onto those of the target client before combination. Oracle is constructed by granting FedIndex access to complete information about all domains, including the target domains. Since vanilla MDD, and DANN are CDA models, we adapt these methods into the FL scenarios based on FedAvg (McMahan et al., 2017) and FADE (Hong et al., 2021) (*i.e.*, FedAvg+MDD, FedAvg+DANN, and FADE+DANN). Besides, we include the naive federated adaptation of CIDA (FedAvg+CIDA) for comparison. For fairness, the network structures of baselines align with FedIndex when components have identical roles.

**Metric:** We employ *accuracy* to evaluate the predictions of FedIndex and baselines on *Half-Circle*, *Sine*, *RotatingMNIST*, *CompCar*, *BrainStroke*, and *TCGA-BRCA*. *Mean squared error (MSE)* loss is used for the *TPT-48* dataset since the task for *TPT-48* is a regression task.

**Non-IID:** We evaluated FedIndex and baselines on two non-independent and identically distributed (IID) settings:

- Feature Shift: Given $X$ domains, $N$ source clients, and one target client, we allocate unique $\lfloor \frac{X}{N+1} \rfloor$ domains to each client sequentially based on their client indices.
- Joint Distribution Shift: We modify the label distribution of clients to simulate a label shift, generating a more complex non-IID setting by combining feature and label shifts.

## 6.2 Results on Toy Datasets

We begin with two toy datasets, *Half-Circle* and *Sine*, to investigate the differences between FedIndex and the baselines under the feature and joint distribution shifts.

**Half-Circle Dataset** comprises 30 domains indexed from 1 to 30. We set $\epsilon = 1$, $\lambda = 0.3$, $N = 4$ and $T = 5$. Hence, we allocate $\lfloor \frac{30}{4+1} \rfloor = 6$ domains to each client sequentially based on the client index (*i.e.*, Clients $\{C_i\}_{i \in [T]}$ are assigned domains indexed from $6 \times i - 5$ to $6 \times i$ respectively).

Table 1: Efficiency of FedIndex on training datasets under the feature distribution shifts (best result in **bold**, second-best underlined). Due to different types of tasks for datasets, we employ *accuracy* for classification tasks on *Half-Circle*, *Sine*, *RotatingMNIST*, *CompCar*, and *BrainStroke*, and *MSE* loss for the regression task on *TPT-48*.

| Model | Half-Circle Accuracy (%) ↑ | Sine Accuracy (%)↑ | RotatingMNIST Accuracy (%)↑ | CompCar Accuracy (%)↑ | BrainStroke Accuracy (%)↑ | TCGA-BRCA Accuracy (%)↑ | TPT-48 MSE ↓ |
|---|---|---|---|---|---|---|---|
| FedAvg+MDD | $48.33 \pm 0.42$ | $78.50 \pm 0.33$ | $9.90 \pm 0.11$ | $34.40 \pm 2.58$ | $42.88 \pm 0.20$ | $73.42 \pm 1.14$ | $2.30 \pm 0.05$ |
| FedAvg+DANN | $80.17 \pm 9.71$ | $\underline{91.40} \pm 0.21$ | $54.30 \pm 3.45$ | $45.09 \pm 3.62$ | $43.83 \pm 0.29$ | $78.90 \pm 1.22$ | $0.45 \pm 0.02$ |
| FADE+DANN | $51.67 \pm 0.64$ | $48.78 \pm 0.31$ | $9.90 \pm 0.13$ | $35.19 \pm 3.50$ | $42.88 \pm 0.22$ | $78.45 \pm 2.11$ | $2.30 \pm 0.06$ |
| FedDA | $48.33 \pm 0.40$ | $48.78 \pm 0.34$ | $45.50 \pm 4.52$ | $5.00 \pm 0.09$ | $42.88 \pm 0.26$ | $73.46 \pm 3.41$ | $\gg 5$ |
| FedGP | $48.33 \pm 0.37$ | $48.85 \pm 0.35$ | $9.90 \pm 0.10$ | $5.00 \pm 0.07$ | $60.43 \pm 3.68$ | $84.55 \pm 2.34$ | $\gg 5$ |
| FedAvg+CIDA | $81.17 \pm 6.93$ | $89.28 \pm 0.27$ | $83.28 \pm 3.55$ | $51.67 \pm 4.60$ | $62.88 \pm 3.24$ | $81.92 \pm 2.74$ | $0.42 \pm 0.02$ |
| **FedIndex** | $\underline{87.80} \pm 7.11$ | $\mathbf{91.50} \pm 0.14$ | $\underline{87.40} \pm 2.48$ | $\underline{53.06} \pm 4.14$ | $\underline{65.57} \pm 3.61$ | $\underline{86.49} \pm 2.33$ | $\underline{0.40} \pm 0.02$ |
| CIDA | $\mathbf{93.72} \pm 0.50$ | $89.04 \pm 0.12$ | $\mathbf{89.30} \pm 2.41$ | $\mathbf{53.66} \pm 1.45$ | $\mathbf{66.99} \pm 3.23$ | $\mathbf{90.68} \pm 1.38$ | $\mathbf{0.37} \pm 0.01$ |
| Oracle | $98.50 \pm 1.10$ | $92.52 \pm 0.10$ | $98.0 \pm 1.71$ | $97.36 \pm 1.74$ | $71.01 \pm 1.09$ | $93.45 \pm 0.90$ | $0.39 \pm 0.01$ |

Table 2: Efficiency of FedIndex on training datasets under joint distribution shift (best result in **bold**, second-best underlined).

| Model | Half-Circle Accuracy (%)↑ | Sine Accuracy (%)↑ | RotatingMNIST Accuracy (%)↑ | CompCar Accuracy (%)↑ | BrainStroke Accuracy (%)↑ | TCGA-BRCA Accuracy (%)↑ | TPT-48 MSE ↓ |
|---|---|---|---|---|---|---|---|
| FedAvg+MDD | $76.99 \pm 3.42$ | $64.96 \pm 0.55$ | $30.90 \pm 0.31$ | $13.47 \pm 0.29$ | $42.88 \pm 0.20$ | $68.2 \pm 3.01$ | $2.78 \pm 0.05$ |
| FedAvg+DANN | $61.50 \pm 0.60$ | $65.05 \pm 0.44$ | $30.90 \pm 0.30$ | $11.94 \pm 0.34$ | $42.90 \pm 0.21$ | $71.55 \pm 2.88$ | $1.17 \pm 0.04$ |
| FADE+DANN | $61.66 \pm 0.64$ | $65.05 \pm 0.53$ | $1.01 \pm 0.14$ | $17.22 \pm 0.41$ | $42.97 \pm 0.18$ | $74.13 \pm 2.52$ | $2.78 \pm 0.06$ |
| FedDA | $38.33 \pm 0.40$ | $34.72 \pm 0.60$ | $29.90 \pm 0.36$ | $9.19 \pm 0.33$ | $42.88 \pm 0.24$ | $65.11 \pm 3.77$ | $\gg 5$ |
| FedGP | $38.33 \pm 0.37$ | $34.72 \pm 0.29$ | $29.90 \pm 0.32$ | $9.19 \pm 0.30$ | $42.88 \pm 0.21$ | $78.18 \pm 2.98$ | $\gg 5$ |
| FedAvg+CIDA | $72.00 \pm 5.45$ | $71.96 \pm 0.51$ | $70.90 \pm 5.35$ | $41.50 \pm 4.48$ | $52.34 \pm 4.22$ | $75.66 \pm 4.82$ | $1.12 \pm 0.04$ |
| **FedIndex** | $\underline{80.50} \pm 4.50$ | $\underline{77.55} \pm 0.62$ | $\underline{83.39} \pm 4.48$ | $\underline{45.57} \pm 4.51$ | $\mathbf{57.97} \pm 3.31$ | $\underline{83.68} \pm 3.12$ | $\underline{0.82} \pm 0.02$ |
| CIDA | $\mathbf{85.10} \pm 2.31$ | $\mathbf{77.96} \pm 0.58$ | $\mathbf{84.50} \pm 4.42$ | $\mathbf{50.22} \pm 2.44$ | $\underline{61.72} \pm 3.48$ | $\mathbf{86.49} \pm 1.82$ | $\mathbf{0.62} \pm 0.01$ |
| Oracle | $87.99 \pm 1.30$ | $77.68 \pm 0.36$ | $97.20 \pm 3.10$ | $83.61 \pm 2.20$ | $65.35 \pm 1.18$ | $90.45 \pm 1.00$ | $0.58 \pm 0.01$ |

From Table 1, we observe that FedDA and FedGP failed on non-IID scenarios since these methods assume that target data are drawn in an IID manner, which is not a general assumption. Besides, FedAvg+MDD completely failed, FedAvg+DANN reached 80.17%, and FedAvg+CIDA achieved 81.17%, significantly lower than FedIndex (87.8%). In contrast, FedIndex achieved 87.8% accuracy under the feature shift, only falling by 4.98% relative to CIDA. This demonstrates 1) categorical FDA struggles to accurately predict in continuously indexed domains due to not considering the domain index, 2) naive adaptation of CIDA to FL scenario cannot achieve the optimum efficiency, and 3) FedIndex effectively accomplishes domain alignment as anticipated by CIDA.

For the joint distribution shift, we adjust the proportion of data labeled as 0 for each client to [0.15, 0.21, 0.27, 0.33, 0.39, 0.45], respectively, by randomly switching labels from 0 to 1. From Table 2, we found FedIndex (80.5%) still outperformed the highest baseline (72.00%) with an 8.50% improvement and was only 4.60% lower than CIDA.

**Sine Dataset** includes 12 domains. Each domain covers $\frac{1}{6}$ sinusoidal period. We set $\epsilon = 1$, $\lambda = 0.3$, $N = 3$, and $T = 4$, with the last three domains as target domains.

Table 1 shows the prediction accuracy of all models under the feature shift. Achieving precise predictions is challenging for all models due to the sine function's oscillatory nature, which leads to similar fluctuations in the model's decision boundary. Although FedIndex provides a more accurate prediction (91.5%) than other baselines, data scarcity makes this difficult to achieve, yielding only a 0.1% improvement over FedAvg+DANN (91.4%). However, in experiments involving a joint distribution shift (adjusting the proportion of 0-labeled samples on each client from 0.5 to [0.25, 0.29, 0.33, 0.37]), FedIndex demonstrates robustness to more complex domain-shift scenarios. Table 2 shows that only FedIndex (77.55%) and CIDA (77.96%) perform effectively

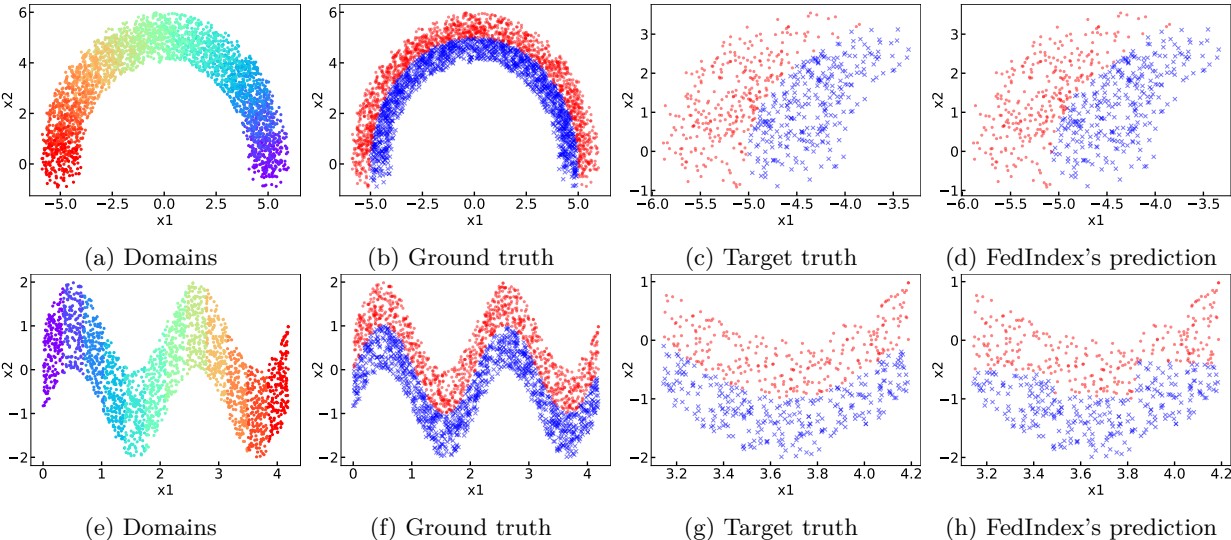

Figure 3: Results on the *Half-circle* and *Sine* datasets, which contain 30 and 12 domains, respectively. Panels a and e visualize the domains via different colors; the last six (three) domains on the left (right) are the target domains for *Half-circle* (*Sine*). Panels b and f show the ground truth for all domains, while c and g show the ground truth restricted to target domains. Panels d and h report FedIndex's predictions.

under the joint distribution shift. In contrast, the best categorical FDA baseline achieves only 65.05% accuracy.

Overall, the results from both the *Half-Circle* and *Sine* datasets demonstrate FedIndex's superiority in capturing the underlying relationship between the domain index and the classification task, thereby enhancing prediction accuracy. In contrast, baselines such as FedAvg+MDD, FedAvg+DANN, and FADE+DANN ignore this underlying relationship, resulting in poorer performance.

### 6.3   Results on RotatingMNIST

We further evaluate different methods on *RotatingMNIST*, which continuously rotates MNIST datasets from $0°$ to $180°$. The domain indices of RotatingMNIST are the rotation angles of the samples. We set $\epsilon = 1$, $\lambda = 1$, $N = 2$ and $T = 3$.

Table 1 presents the superiority of FedIndex under the feature shift. FedIndex (87.4%) only fails 1.9% of CIDA (89.3%). In contrast, other baselines that ignore the underlying relations between domain indices and the task cannot work efficiently on RotatingMNIST. The best baseline, FedAvg+DANN, obtains a 54.3% accuracy, a 36.1% gap compared to FedIndex. The main reason resulting in sub-optimal accuracy of baselines is that digit pairs (6, 9) and (2, 7) are hard to distinguish after rotation.

For joint distribution shift, we use a Dirichlet distribution with $\alpha = 0.5$ to introduce label shift alongside feature shift. In this complex non-IID setting, all models, except the Oracle, suffer accuracy degradation. For example, FedIndex shows a 4% drop compared to its performance under feature shift. Nevertheless, Table 2 shows a consistent accuracy trend, and FedIndex still achieves a 52.4% improvement over the best FDA baseline, demonstrating strong robustness in challenging scenarios.

### 6.4   Results on CompCar

In *CompCar*, data from each viewpoint and each YOM is treated as an individual domain. We use the dataset processed by VDI (Xu et al., 2023), which has four car types (MPV, SUV, sedan, and hatchback) and five viewpoints (front (F), rear (R), side (S), front-side (FS), and rear-side (RS)), spanning model years from 2009 to 2014. The entire dataset has 30 domains (5 viewpoints × 6 YOM) with 18,735 images.

Table 3: Prediction accuracy (%) of FedIndex on *TCGA-BRCA* (feature distribution shift) when varying the target age domain. For each experiment, one age range is used as the target domain, and all remaining ranges serve as source domains.

| Model | [25,34] | [35,44] | [45,54] | [55,64] | [65,74] | [75,84] |
|---|---|---|---|---|---|---|
| FedIndex | $82.2 \pm 2.45$ | $83.91 \pm 2.35$ | $87.41 \pm 2.51$ | $89.77 \pm 2.17$ | $87.53 \pm 2.41$ | $86.49 \pm 2.33$ |

We evaluate *CompCar* based on the *accuracy* of predicting the car type from an image under the settings ($\epsilon = 1$, $\lambda = 1$, $N = 5$, $T = 6$). For feature shift, Table 1 demonstrates that FedIndex achieves an accuracy of 53.06%, reflecting a 7.97% improvement over the best FDA baseline (45.09%) and comparable performance to CIDA.

For joint distribution shift, we use a Dirichlet distribution with $\alpha = 0.8$ to simulate label shift and combine it with feature shift. As shown in Table 2, the complex non-IID settings result in significant performance degradation across all baselines, whereas FedIndex maintains a robust performance with an accuracy of 45.57%.

## 6.5 Results on TCGA-BRCA

We use the publicly available dataset (The Cancer Genome Atlas Network, 2012), *TCGA-BRCA*, to further assess the effectiveness of FedIndex on large-scale, real-world genomic data. *TCGA-BRCA* contains comprehensive molecular and clinical profiles of breast cancer patients collected as part of The Cancer Genome Atlas program. *TCGA-BRCA* contains 1095 samples, where each sample includes: (1) over 18000 high-dimensional gene expression features (RNA-Seq), (2) 39 clinical features (*e.g.*, tumor stage, survival time, receptor statuses, etc.), (3) binary subtype labels distinguishing Basal-like versus non-Basal tumors, and (4) demographic and clinical covariates used as domain indices (*e.g.*, age group). Specifically, we build six domains, each containing samples from 10 consecutive years, ranging from 25 to 84 years old (*i.e.*, [25,34], [35,44], [45,54], [55,64], [65,74], [75,84]). We set $\epsilon = 1$, $\lambda = 1$, $N = 5$, and $T = 6$ for all experiments conducted on this dataset.

Under feature shift conditions, as shown in Table 1, FedIndex achieves an accuracy of 86.49%, outperforming the best FDA baseline (84.55% from FedGP). This performance gain highlights FedIndex's ability to leverage continuous domain indices to model gradual genomic variation across age groups. Since breast cancer molecular subtypes exhibit well-documented age-dependent patterns, incorporating age as a continuous variable enables FedIndex to better capture the underlying structure that categorical FDA approaches fail to represent.

To evaluate joint distribution shifts, we use a Dirichlet distribution with $\alpha = 0.5$ to simulate label shift and combine it with feature shift. Feature shifts are then applied simultaneously, forming a challenging joint feature-and-label shift scenario. As shown in Table 2, only FedIndex maintains a high accuracy of 83.68%, whereas all other baselines, including the strongest categorical FDA method, suffer substantial performance drops. These results underscore the robustness of FedIndex in modeling smooth domain transitions and handling non-IID genomic data, demonstrating its practical value for large-scale, privacy-sensitive medical datasets such as *TCGA-BRCA*.

Furthermore, to examine the effect of domain selection, we varied the target domain by sequentially designating each age range as the held-out target. Table 3 reports the results across different target choices. We observe that FedIndex achieves higher accuracy on target domains corresponding to middle-aged groups (*e.g.*, [45,54] and [55,64]) compared to domains at the extremes (*e.g.*, [25,34] and [75,84]). This pattern further suggests that age, as a continuous domain index, captures meaningful correlations with the prediction task: when the target domain lies in the middle of the index range, FedIndex effectively leverages information from both neighboring age groups (*e.g.*, [25,44] and [55,84]) and learns a smooth interpolation, leading to improved prediction performance.

### 6.6 Results on BrainStroke

We utilize the publicly available dataset (Soriano, 2021), referred to as *BrainStroke*, which comprises 5,510 samples to evaluate the efficiency of FedIndex on real-world medical datasets. In *BrainStroke*, each sample includes: 1) features related to brain stroke prediction (*e.g.*, gender, hypertension, heart disease, average glucose level, BMI, and smoking status), 2) binary labels for brain stroke classification (1 indicating stroke and 0 indicating no stroke), and 3) domain indices (*e.g.*, age). We set $\epsilon = 1$, $\lambda = 1$, $N = 2$ and $T = 3$.

Under feature shift conditions, as shown in Table 1, FedIndex achieves a prediction accuracy of 65.57%, failing 1.42% of CIDA (66.99%) and outperforming the best FDA baseline (60.43%). The accuracy improvement of FedIndex over the baselines arises from leveraging the continuous domain indices. Since older individuals are more susceptible to brain strokes, age serves as a continuous domain index that captures the underlying relationships relevant to the task.

To evaluate joint distribution shifts, we first simulate label shifts by altering a subset of samples within each domain, flipping labels from 0 to 1. The probability of label flipping increases with the domain indices (*i.e.*, higher probabilities for older individuals). We then introduce feature shifts alongside these label shifts to generate a combined joint distribution shift. In such challenging scenarios, Table 2 shows that only FedIndex maintains a prediction accuracy of 57.97%, whereas all other baselines, focusing on categorical domain adaptation, fail to perform in this complex non-IID setting. These results highlight the importance of continuous indices and demonstrate FedIndex's robustness to domain shifts and data scarcity, especially in privacy-preserving real-world medical datasets.

### 6.7 Results on TPT-48

*TPT-48* contains monthly average temperatures for the 48 contiguous states in the US from 2008 to 2019. The raw data are from the National Oceanic and Atmospheric Administration's Climate Divisional Database (nClimDiv) and Gridded 5km GHCN-Daily Temperature and Precipitation Dataset (nClimGrid) (Vose et al., 2015). We use the data processed by Washington Post (Post, 2019) and GRDA (Xu et al., 2022).

Here, we evaluate FedIndex on the regression task to predict the next six months' temperature based on the previous six months' temperature with the setting ($\epsilon = 1$, $\lambda = 1$, $N = 2$, and $T = 3$). Clients' domains are sequentially indexed from North to South (Xu et al., 2022). For the feature shift, Table 1 shows FedIndex achieves MSE loss 0.4, better than that of the best FDA baseline, 0.45.

For the joint distribution shift, we scale the temperatures of clients' months with coefficients $[0.3, 0.5, 0.7]$ to generate the label shift. Table 2 presents that FedIndex still captures the underlying relation between domain indices and the task, resulting in lower MSE loss and fewer rounds for convergence.

### 6.8 Results on Standard Domain Adaptation Benchmarks

To further validate the robustness and scalability of FedIndex beyond the synthetic toy datasets, we evaluate its performance on several large-scale and complex DA benchmarks: *Office-Home* (Venkateswara et al., 2017), *DomainNet* (Peng et al., 2019a), *DG-15* (Xu et al., 2022), and *Caltech Camera Traps (CCT)* (Beery et al., 2018). By including the rotating versions of *Office-Home* and *DomainNet*, *i.e.*, *RotatingOffice-Home* and *RotatingDomainNet*, we specifically examine how FedIndex handles continuous domain shifts in a federated environment, which is a critical requirement for practical deployments where environment changes are often gradual rather than discrete. For all benchmarks, we use a Dirichlet distribution with $\alpha = 0.5$ to simulate label shift and combine it with feature shift to model joint distribution shift.

**Office-Home.**   *Office-Home* contains approximately 15,500 images spanning 65 object categories and 4 distinct domains (*Art, Clipart, Product, Real-World*). Each sample can be represented as a triplet $(x, y, u)$, where $x \in \mathbb{R}^{224 \times 224 \times 3}$ denotes an RGB image, $y \in \{1, \ldots, 65\}$ is the object category label, and $u \in \{1, \ldots, 4\}$ indicates the domain index corresponding to its visual style (e.g., *Art* or *Product*). We set $\epsilon = 1$, $\lambda = 1$, $N = 3$, and $T = 4$, with each client assigned to a specific domain. The domains exhibit significant feature shifts

Table 4: Efficiency of FedIndex on DA benchmarks under the feature distribution shifts (best result in **bold**, second-best underlined). We employ *accuracy (%)* for classification tasks on *Office-Home*, *RotatingOffice-Home*, *DomainNet*, *RotatingDomainNet*, *DG-15*, and *Caltech Camera Traps (CCT)*.

| Model | Office-Home Accuracy (%) ↑ | Rot. Office-Home Accuracy (%) ↑ | DomainNet Accuracy (%) ↑ | Rot. DomainNet Accuracy (%) ↑ | DG-15 Accuracy (%) ↑ | CCT Accuracy (%) ↑ |
|---|---|---|---|---|---|---|
| FedAvg+MDD | $54.23 \pm 2.13$ | $41.87 \pm 2.88$ | $21.83 \pm 1.72$ | $24.92 \pm 2.31$ | $49.78 \pm 1.15$ | $58.63 \pm 2.04$ |
| FedAvg+DANN | $60.82 \pm 2.34$ | $53.68 \pm 2.46$ | $26.93 \pm 1.68$ | $24.43 \pm 2.11$ | $71.87 \pm 1.21$ | $65.42 \pm 1.87$ |
| FADE+DANN | $57.12 \pm 2.51$ | $45.58 \pm 2.73$ | $23.74 \pm 1.95$ | $21.63 \pm 2.40$ | $67.53 \pm 1.44$ | $62.28 \pm 2.05$ |
| FedDA | $52.57 \pm 2.22$ | $43.13 \pm 2.31$ | $19.88 \pm 1.54$ | $18.34 \pm 1.98$ | $58.37 \pm 1.37$ | $55.82 \pm 1.74$ |
| FedGP | $61.43 \pm 2.18$ | $49.78 \pm 2.27$ | $28.64 \pm 1.63$ | $33.17 \pm 2.06$ | $69.83 \pm 1.26$ | $66.18 \pm 1.68$ |
| FedAvg+CIDA | $65.93 \pm 2.40$ | $68.72 \pm 2.12$ | $35.94 \pm 1.71$ | $38.83 \pm 1.84$ | $79.57 \pm 1.08$ | $72.84 \pm 1.56$ |
| **FedIndex** | $\underline{69.12} \pm 2.03$ | $\underline{72.58} \pm 1.88$ | $\underline{39.34} \pm 1.42$ | $\underline{43.87} \pm 1.61$ | $\underline{83.23} \pm 0.96$ | $\underline{76.12} \pm 1.32$ |
| CIDA | $\mathbf{70.84} \pm 1.74$ | $\mathbf{74.88} \pm 1.63$ | $\mathbf{42.83} \pm 1.35$ | $\mathbf{45.12} \pm 1.49$ | $\mathbf{85.04} \pm 0.81$ | $\mathbf{77.43} \pm 1.10$ |
| Oracle | $82.34 \pm 1.12$ | $80.08 \pm 1.06$ | $58.64 \pm 1.08$ | $62.43 \pm 1.17$ | $88.12 \pm 0.62$ | $81.57 \pm 0.94$ |

Table 5: Efficiency of FedIndex on DA benchmarks under the joint distribution shifts (best result in **bold**, second-best underlined). We employ *accuracy (%)* for classification tasks on *Office-Home*, *RotatingOffice-Home*, *DomainNet*, *RotatingDomainNet*, *DG-15*, and *Caltech Camera Traps (CCT)*.

| Model | Office-Home Accuracy (%) ↑ | Rot. Office-Home Accuracy (%) ↑ | DomainNet Accuracy (%) ↑ | Rot. DomainNet Accuracy (%) ↑ | DG-15 Accuracy (%) ↑ | CCT Accuracy (%) ↑ |
|---|---|---|---|---|---|---|
| FedAvg+MDD | $49.57 \pm 2.36$ | $37.18 \pm 3.07$ | $18.73 \pm 1.95$ | $21.14 \pm 2.56$ | $45.38 \pm 1.32$ | $53.27 \pm 2.28$ |
| FedAvg+DANN | $56.48 \pm 2.58$ | $48.23 \pm 2.69$ | $23.08 \pm 1.89$ | $21.07 \pm 2.34$ | $67.63 \pm 1.39$ | $61.18 \pm 2.02$ |
| FADE+DANN | $52.93 \pm 2.73$ | $40.37 \pm 2.95$ | $19.87 \pm 2.14$ | $18.12 \pm 2.61$ | $63.58 \pm 1.63$ | $58.13 \pm 2.22$ |
| FedDA | $48.31 \pm 2.41$ | $38.34 \pm 2.52$ | $16.74 \pm 1.71$ | $15.57 \pm 2.12$ | $54.03 \pm 1.55$ | $51.43 \pm 1.96$ |
| FedGP | $58.04 \pm 2.43$ | $46.02 \pm 2.61$ | $25.93 \pm 1.82$ | $29.17 \pm 2.29$ | $66.12 \pm 1.43$ | $62.04 \pm 1.91$ |
| FedAvg+CIDA | $62.87 \pm 2.61$ | $65.54 \pm 2.34$ | $32.43 \pm 1.89$ | $35.37 \pm 2.03$ | $75.92 \pm 1.21$ | $69.57 \pm 1.73$ |
| **FedIndex** | $\underline{66.08} \pm 2.22$ | $\underline{69.24} \pm 2.05$ | $\underline{35.87} \pm 1.58$ | $\underline{40.34} \pm 1.79$ | $\underline{79.03} \pm 1.08$ | $\underline{73.22} \pm 1.48$ |
| CIDA | $\mathbf{67.92} \pm 1.96$ | $\mathbf{71.13} \pm 1.82$ | $\mathbf{38.68} \pm 1.47$ | $\mathbf{41.94} \pm 1.63$ | $\mathbf{80.74} \pm 0.94$ | $\mathbf{74.44} \pm 1.26$ |
| Oracle | $79.28 \pm 1.27$ | $77.23 \pm 1.18$ | $54.97 \pm 1.19$ | $58.43 \pm 1.31$ | $85.58 \pm 0.76$ | $79.17 \pm 1.05$ |

while sharing the same label space, making the dataset well-suited for evaluating cross-domain generalization and client heterogeneity in federated settings.

On *Office-Home*, FedIndex achieves 69.12% accuracy under feature shift and 66.08% under joint distribution shift. Compared with the strongest federated baseline, *FedAvg+CIDA*, FedIndex improves performance by 3.19% and 3.21%, respectively. Moreover, FedIndex remains close to centralized *CIDA*, with gaps of only 1.72% under feature shift and 1.84% under joint distribution shift. These results indicate that, even on a standard discrete-domain benchmark, the proposed global discriminator enables more effective cross-client alignment than both categorical FDA baselines and the naive federated adaptation of *CIDA*.

***RotatingOffice-Home.*** To further evaluate the capability of handling continuous domain shifts, we construct a synthetic variant, *RotatingOffice-Home*, inspired by RotatedMNIST. Specifically, each image $x$ is transformed via a rotation operator $T_\theta(x)$, where $\theta \in [0°, 360°)$ is a continuous domain index representing the rotation angle. Accordingly, each sample is represented as $(x_\theta, y, u_\theta)$, where $x_\theta = T_\theta(x)$ and $u_\theta = \theta$ denotes a continuous domain variable rather than a discrete index in standard *Office-Home*, *e.g.*, *Product*. The rotation angles are assigned to simulate continual domain evolution across clients by uniformly partitioning the angle space based on the number of clients. Specifically, for a setup with $T$ total clients, the $i$-th client $C_i$ is associated with a rotation angle $\theta_i = \frac{360 \times i}{T}$, and contains samples transformed by $T_{\theta_i}(\cdot)$. Unless otherwise specified, we set $\epsilon = 1$, $\lambda = 1$, $N = 5$, and $T = 6$. This construction introduces smooth and continuous feature shifts while preserving label semantics, thereby enabling a more fine-grained evaluation of a model's ability to generalize under evolving distributions.

On *RotatingOffice-Home*, FedIndex achieves 72.58% accuracy under feature shift and 69.24% under joint distribution shift, outperforming *FedAvg+CIDA* by 3.86% and 3.70%, respectively. The gains over categorical

FDA baselines are even larger, e.g., 18.90% over *FedAvg+DANN* under feature shift and 21.01% under joint distribution shift. Meanwhile, FedIndex remains within 2.30% and 1.89% of centralized *CIDA* in the two settings. These results suggest that the advantage of FedIndex becomes more pronounced when domain variation is continuous and evolves smoothly across clients, which is precisely the setting that motivates continuous domain indexing.

***DomainNet.*** *DomainNet* is a large-scale domain generalization benchmark containing approximately 0.6 million images across 345 categories and 6 domains (*Clipart, Infograph, Painting, Quickdraw, Real, Sketch*). Each sample can be represented as a triplet $(x, y, u)$, where $x \in \mathbb{R}^{H \times W \times 3}$ denotes an RGB image, $y \in \{1, \ldots, 345\}$ is the object category label, and $u \in \{1, \ldots, 6\}$ indicates the domain index corresponding to its visual style. Following standard practice, all images are resized and center-cropped to $224 \times 224$, *i.e.*, $H = W = 224$. Here, we set $\epsilon = 1$, $\lambda = 1$, $N = 5$, and $T = 6$. The dataset exhibits substantial feature shifts across domains while sharing a common label space, making it a challenging benchmark for evaluating scalability under domain heterogeneity.

On the large-scale *DomainNet* benchmark, FedIndex reaches 39.34% accuracy under feature shift and 35.87% under joint distribution shift. Although the benchmark is substantially more challenging due to its larger number of classes and stronger heterogeneity, FedIndex still consistently outperforms *FedAvg+CIDA* by 3.40% and 3.44%, respectively. In addition, FedIndex improves markedly over categorical FDA baselines such as *FedAvg+DANN*, with gains of 12.41% under feature shift and 12.79% under joint distribution shift. This demonstrates that the adversarial interaction between local encoders and the centralized global discriminator scales effectively to complex and large-domain adaptation scenarios.

***RotatingDomainNet.*** To further evaluate robustness under continuous domain shifts at scale, we construct *RotatingDomainNet* by introducing rotation-based transformations. The rotation operation in *RotatingDomainNet* is the same as that in *RotatingOffice-Home* and *RotatingMNIST*. Here, we set $\epsilon = 1$, $\lambda = 1$, $N = 5$, and $T = 6$.

Under this setting, FedIndex attains 43.87% accuracy under feature shift and 40.34% under joint distribution shift. Compared with *FedAvg+CIDA*, the performance gains further increase to 5.04% and 4.97%, respectively, which are larger than those observed on the non-rotating *DomainNet*. Relative to *FedAvg+DANN*, the gains are even more substantial, reaching 19.44% under feature shift and 19.27% under joint distribution shift. These results further verify that continuous domain indexing is particularly beneficial when the domain shift is smooth and continuously varying, rather than merely categorical.

***DG-15.*** *DG-15* is a synthetic binary classification benchmark consisting of 15 domains with structured inter-domain relationships. Each sample is represented as $(x, y, u)$, where $x \in \mathbb{R}^2$ denotes a 2-dimensional feature vector, $y \in \{0, 1\}$ is the class label, and $u \in \{1, \ldots, 15\}$ indicates the domain index. In each domain $i$, data are generated from two Gaussian distributions $\mathcal{N}(\mu_{i,1}, \mathbf{I})$ and $\mathcal{N}(\mu_{i,0}, \mathbf{I})$ corresponding to the positive and negative classes, respectively, where the means are determined by a domain-specific embedding. Specifically, each domain is associated with a unit vector $[a_i, b_i]$, which defines its position in the embedding space and controls the orientation of its decision boundary. Here, we set $\epsilon = 1$, $\lambda = 1$, $N = 4$, and $T = 5$. As a result, domains with similar embeddings exhibit similar class distributions and decision boundaries.

FedIndex achieves 83.23% accuracy under feature shift and 79.03% under joint distribution shift. It consistently outperforms *FedAvg+CIDA* by 3.66% and 3.11%, respectively, and exceeds *FedAvg+DANN* by 11.36% and 11.40%. Furthermore, FedIndex remains close to centralized *CIDA*, with only 1.81% and 1.71% performance gaps in the two settings. Since *DG-15* contains structured inter-domain relationships with smoothly varying decision boundaries, these results indicate that FedIndex can effectively exploit such structured domain geometry in federated settings.

***CCT.*** *CCT* is a large-scale wildlife monitoring dataset consisting of 57,864 images collected from 20 camera locations. Each sample is represented as $(x, y, u)$, where $x \in \mathbb{R}^{H \times W \times 3}$ denotes an RGB image, $y \in \{0, 1\}$ indicates the presence or absence of animals, and $u$ corresponds to the camera location index. The dataset exhibits substantial domain shifts due to variations in geographic location, illumination conditions (*e.g.*, day vs. night), background clutter, and camera viewpoints. Here, we set $\epsilon = 1$, $\lambda = 1$, $N = 4$, and $T = 5$.

On *CCT*, FedIndex obtains 76.12% accuracy under feature shift and 73.12% under joint distribution shift. Compared with *FedAvg+CIDA*, this corresponds to gains of 3.28% and 3.55%, respectively, while the gains over *FedAvg+DANN* are 10.70% and 11.94%. Moreover, FedIndex remains close to the centralized *CIDA*, with a gap of only 1.31% across both settings. These results show that FedIndex generalizes well not only to synthetic or benchmark-style domain shifts but also to real-world environmental shifts induced by camera location, illumination, and background variation.

Across all DA benchmarks in Tables 4 and 5, FedIndex is consistently the strongest federated method under both feature shift and joint distribution shift. A clear pattern is that FedIndex's improvement over existing federated baselines is especially pronounced on the rotating benchmarks, where domain evolution is continuous by construction. This supports our central claim that explicitly modeling domain indices as continuous variables is more effective than treating domains as isolated categorical labels. At the same time, the consistent improvement of FedIndex over *FedAvg+CIDA* across all datasets confirms that naively federating a centralized continuous DA method remains suboptimal in FL due to the lack of a globally trained discriminator. Finally, the fact that FedIndex remains consistently close to centralized *CIDA* shows that the proposed design successfully bridges much of the gap between centralized continuous DA and privacy-preserving federated training, while maintaining robustness across benchmarks of different scales, modalities, and domain structures.

### 6.9 Performance Gap between FedIndex and Oracle

Although FedIndex substantially outperforms existing FDA baselines, there remains a performance gap between FedIndex and Oracle on some datasets. For example, on RotatingMNIST, FedIndex achieves strong performance compared to prior FDA baselines but still falls behind the Oracle. This gap is expected because Oracle is not a deployable FDA method; rather, it is a federated upper-bound baseline. Specifically, Oracle follows the same federated training protocol and uses the same model architecture as FedIndex, but it is provided with additional target-domain information unavailable to FedIndex in practical FDA settings. Such a gap reflects the costs of realistic information constraints, privacy-preserving perturbations, residual domain discrepancies, and non-convex federated minimax optimization.

Formally, let $h_{\text{FedIndex}}$ denote the model learned by FedIndex and $h_{\text{Oracle}}$ denote the model learned by the federated Oracle baseline. Both methods are trained under the same federated protocol. Let $R_T(h)$ be the target-domain risk of a hypothesis $h$. The gap between FedIndex and Oracle can be decomposed as

$$R_T(h_{\text{FedIndex}}) - R_T(h_{\text{Oracle}}) \leq \mathcal{E}_{\text{info}} + \mathcal{E}_{\text{align}} + \mathcal{E}_{\text{DP}} + \mathcal{E}_{\text{opt}},$$

where $\mathcal{E}_{\text{info}}$ captures the cost of unavailable target-domain information, $\mathcal{E}_{\text{align}}$ captures residual discrepancy after adversarial domain alignment, $\mathcal{E}_{\text{DP}}$ captures the finite-sample effect of privacy-preserving perturbation, and $\mathcal{E}_{\text{opt}}$ captures the optimization error induced by the non-convex federated min-max objective.

**Information gap.** The main difference between FedIndex and Oracle lies in the information available during training. Oracle has access to additional target-domain information, while FedIndex must infer the target-domain structure indirectly from source supervision, continuous domain indices, and adversarial alignment. This creates an information gap:

$$\mathcal{E}_{\text{info}} = \lambda^{\star}_{\text{FedIndex}} - \lambda^{\star}_{\text{Oracle}},$$

where

$$\lambda^{\star}_{\text{Fed}} = \min_{h \in \mathcal{H}} \left[ R_S(h) + R_T(h) \right]$$

denotes the best joint source-target risk achievable under the information available to FedIndex, and $\lambda^{\star}_{\text{Oracle}}$ denotes the corresponding joint optimal risk when the additional oracle information is available. Since Oracle has strictly more information than FedIndex, we generally have

$$\lambda^{\star}_{\text{Oracle}} \leq \lambda^{\star}_{\text{FedIndex}}.$$

Therefore, $\mathcal{E}_{\text{info}}$ is generally nonzero. This term explains why FedIndex cannot always match Oracle even when its domain alignment is effective.

**Residual alignment gap.** FedIndex aligns continuously indexed domains through the global discriminator. While our framework supports the alignment of higher-order moments, the current MSE-based discriminator objective mainly enforces first-moment alignment of the continuous domain index at equilibrium. This does not necessarily imply full distributional alignment between source and target feature distributions. Therefore, a residual discrepancy may remain:

$$\mathcal{E}_{\text{align}} = d_{\mathcal{H}}\big(P_S^{Z_{\text{FedIndex}}}, P_T^{Z_{\text{FedIndex}}}\big) - d_{\mathcal{H}}\big(P_S^{Z_{\text{Oracle}}}, P_T^{Z_{\text{Oracle}}}\big),$$

where $P_S^Z$ and $P_T^Z$ denote the source and target feature distributions after encoding, and $d_{\mathcal{H}}(\cdot, \cdot)$ is a hypothesis-induced discrepancy measure. Oracle can reduce this discrepancy more effectively because it has access to additional target-domain information. In contrast, FedIndex must reduce the discrepancy only through adversarial signals from noisy embeddings and continuous domain indices.

**Privacy-induced gap.** FedIndex applies Laplace perturbation to transmitted embeddings and domain indices for privacy preservation. Although our theoretical analysis shows that zero-mean Laplace noise does not shift the optimal alignment condition at equilibrium, it can still affect finite-sample training by perturbing the discriminator signal. Let

$$\tilde{z} = z + \tau_z, \qquad \tilde{u} = u + \tau_u,$$

where $\tau_z$ and $\tau_u$ are Laplace noises with scale $b = \Delta f / \epsilon$. If the discriminator loss is Lipschitz with respect to the embedding and domain index, then

$$\mathbb{E}\left[|L_d(D(z + \tau_z), u + \tau_u) - L_d(D(z), u)|\right] \leq C_z \mathbb{E}\|\tau_z\| + C_u \mathbb{E}|\tau_u|.$$

Since the magnitude of Laplace noise scales with $b = \Delta f / \epsilon$, the finite-sample privacy-induced gap satisfies

$$\mathcal{E}_{\text{DP}} = O\left(\frac{\Delta f}{\epsilon}\right).$$

Thus, stronger privacy protection, corresponding to smaller $\epsilon$, can enlarge the finite-sample gap to Oracle, even though the equilibrium alignment condition remains unchanged.

**Optimization gap.** Both FedIndex and Oracle are trained under a federated protocol, but FedIndex still solves a non-convex min-max problem involving local encoders, local predictors, and a global discriminator. Therefore, the learned model may not reach the ideal equilibrium characterized by the theoretical analysis. Let $\theta_{\text{FedIndex}}$ denote the parameters obtained by FedIndex and $\theta_{\text{FedIndex}}^{\star}$ denote the ideal equilibrium under the same information constraints. If the target risk is $L$-Lipschitz with respect to model parameters, then

$$\mathcal{E}_{\text{opt}} \leq L \left\|\theta_{\text{FedIndex}} - \theta_{\text{FedIndex}}^{\star}\right\|.$$

This term captures the effect of finite communication rounds, local training drift, client heterogeneity, and the difficulty of optimizing the adversarial objective.

Overall, the FedIndex–Oracle gap reflects the cost of operating under practical FDA constraints rather than a failure to exploit continuous domain indices. This gap can be especially visible on datasets such as RotatingMNIST, where some rotated digit pairs, *e.g.*, $(6, 9)$ and $(2, 7)$, become visually ambiguous. In such cases, additional oracle information can directly reduce target-domain uncertainty, while FedIndex can only infer the target structure through source labels and continuous-domain alignment. Closing this gap further may require stronger alignment objectives beyond first-moment matching, more informative target-side self-training, or adaptive mechanisms for balancing prediction and alignment losses.

### 6.10 Ablation Study

This section evaluates how each component of FedIndex influences prediction accuracy across different datasets.

Table 6: Ablation study of FedIndex on toy, vision, and real-world datasets. "FedIndex" denotes the original model, "w/o DP" removes the DP mechanism, and "w/o index" replaces the continuous domain index with the categorical domain index (*e.g.*, "0 → source" and "1 → target").

| Method | Half-Circle Accuracy (%)↑ | Sine Accuracy (%)↑ | RotatingMNIST Accuracy (%)↑ | CompCar Accuracy (%)↑ | BrainStroke Accuracy (%)↑ | TCGA-BRCA Accuracy (%)↑ | TPT-48 MSE↓ |
|---|---|---|---|---|---|---|---|
| FedIndex | $87.8 \pm 7.11$ | $91.50 \pm 0.14$ | $87.4 \pm 2.48$ | $53.06 \pm 4.14$ | $65.57 \pm 3.61$ | $86.49 \pm 2.33$ | $0.40 \pm 0.02$ |
| w/o DP | $88.9 \pm 4.29$ | $91.70 \pm 0.10$ | $87.95 \pm 1.94$ | $53.50 \pm 3.01$ | $66.75 \pm 4.21$ | $88.97 \pm 1.88$ | $0.39 \pm 0.02$ |
| w/o index | $80.17 \pm 8.54$ | $91.40 \pm 0.33$ | $57.4 \pm 2.54$ | $46.06 \pm 3.87$ | $45.57 \pm 0.61$ | $76.89 \pm 1.32$ | $0.45 \pm 0.02$ |

**With/without DP:** Table 6 compares prediction accuracy with and without DP. The results indicate that additive Laplace noise, parameterized by $b = \frac{\Delta f}{\epsilon}$, simultaneously provides security guarantees and minimally impacts prediction accuracy, consistent with the theoretical analysis in Section 5.

**With/without Continuous Domain Indices:** Table 6 demonstrates that models neglecting continuous domain indices in continuously indexed domains either exhibit lower prediction accuracy or require additional computational resources for convergence.

### 6.11   Selection of $\lambda_d$

The hyperparameter $\lambda_d$ controls the trade-off between task prediction and adversarial domain alignment in Eq. 1. A larger $\lambda_d$ encourages stronger removal of domain-specific information from the learned representation, while a smaller $\lambda_d$ places more emphasis on preserving task-discriminative information. Therefore, $\lambda_d$ should not be chosen solely to maximize domain confusion, especially when the continuous domain index is informative for the prediction task.

In practical FDA settings, Oracle and centralized CIDA validation may be unavailable. Hence, $\lambda_d$ should be selected using only signals that are available under the federated protocol. We propose a practical validation rule based on three quantities: source-client validation performance, cross-client stability, and discriminator-based alignment quality. Specifically, each source client holds out a small labeled validation split and reports only the validation metric to the server. For each candidate $\lambda_d \in \Lambda$, FedIndex is trained for a small number of warm-up rounds, and the server computes the following score:

$$S(\lambda_d) = \bar{A}_{src}(\lambda_d) - \beta \cdot \mathrm{Var}_{i \in [N]} A_i(\lambda_d) - \gamma \cdot \mathcal{D}_{align}(\lambda_d),$$

where $\bar{A}_{src}(\lambda_d)$ is the average source validation accuracy, $A_i(\lambda_d)$ is the validation accuracy on client $i$, $\mathrm{Var}_{i \in [N]} A_i(\lambda_d)$ measures the stability of performance across source clients, and $\mathcal{D}_{align}(\lambda_d)$ measures the residual domain information contained in the learned embeddings. The constants $\beta$ and $\gamma$ control the importance of cross-client stability and alignment.

The alignment term can be estimated by the global discriminator. Since the discriminator is trained to recover the continuous domain index from the embedding, a smaller discriminator prediction error indicates stronger domain leakage, while a larger discriminator prediction error indicates stronger domain confusion. Therefore, one possible choice is

$$\mathcal{D}_{align}(\lambda_d) = -\mathcal{L}_d^{val}(\lambda_d),$$

where $\mathcal{L}_d^{val}(\lambda_d)$ is the discriminator's validation loss for predicting the continuous domain index from the learned embeddings. Minimizing $\mathcal{D}_{align}$ is then equivalent to encouraging a larger discriminator error, *i.e.*, stronger domain alignment.

In practice, we use a conservative selection rule. Let

$$\bar{A}_{src}^{best} = \max_{\lambda_d \in \Lambda} \bar{A}_{src}(\lambda_d), \qquad \mathcal{L}_d^{best} = \max_{\lambda_d \in \Lambda} \mathcal{L}_d^{val}(\lambda_d),$$

where $\mathcal{L}_d^{best}$ corresponds to the strongest observed domain confusion. We choose the smallest $\lambda_d$ that achieves near-optimal domain alignment while preserving source validation performance:

$$\lambda_d^\star = \min \left\{ \lambda_d \in \Lambda : \mathcal{L}_d^{val}(\lambda_d) \geq \mathcal{L}_d^{best} - \delta \quad \text{and} \quad \bar{A}_{src}(\lambda_d) \geq \bar{A}_{src}^{best} - \eta \right\},$$

where $\delta$ and $\eta$ are tolerance parameters. This rule selects the weakest adversarial strength that produces sufficient domain alignment without noticeably sacrificing predictive accuracy. Such a conservative choice is useful because overly large $\lambda_d$ may over-align representations and remove information that is useful for prediction.

This selection rule is consistent with our theoretical analysis in Appendix C.5. When the domain index $u$ is a nuisance variable independent of the label $y$, stronger alignment is beneficial because it removes domain-specific variation without harming task prediction. In this case, increasing $\lambda_d$ can improve target-domain generalization. However, when $u$ is informative for predicting $y$, the prediction objective and the alignment objective conflict: minimizing the prediction loss requires retaining some information associated with $u$, while maximizing domain confusion encourages removing it. Therefore, an excessively large $\lambda_d$ can discard task-relevant information and degrade performance. The proposed validation rule balances these two effects by requiring sufficient alignment while explicitly constraining the degradation of source validation accuracy.

Empirically, this behavior is verified through sensitivity analysis over $\lambda_d$ in Table 9. When $\lambda_d$ is too small, the encoder receives weak adversarial signals, and the learned representation may retain excessive domain-specific information. When $\lambda_d$ is moderate, FedIndex achieves better domain alignment while preserving task-discriminative information, leading to improved target-domain performance. When $\lambda_d$ is too large, the adversarial objective dominates the prediction objective, which can lead to over-alignment and degraded predictive performance. Thus, the best performance is typically achieved in an intermediate range of $\lambda_d$, supporting the proposed practical selection strategy.

# 7 Conclusion

We address the challenge of FDA across domains characterized by continuous domain indices by proposing FedIndex, a federated method explicitly designed to operate under this setting while preserving data privacy. FedIndex adopts a privacy-preserving adversarial learning framework that enables effective incorporation of continuous domain information without requiring access to raw client data or explicit domain boundaries. We provide both theoretical analysis, which characterizes the behavior of FedIndex under continuous domain shifts, and extensive empirical evaluations on synthetic benchmarks and real-world datasets. Experimental results demonstrate the effectiveness and robustness of a general FDA framework that seamlessly integrates continuous domain indices, highlighting its potential for deployment in privacy-sensitive applications, such as healthcare and biomedical modeling. Moreover, FedIndex provides a flexible, extensible foundation for adapting to evolving domain characteristics in federated environments. As future work, we plan to extend this framework to more challenging settings involving domain shift across clients and over time, such as federated continual learning scenarios.

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

## A   Pseudo Code

This section provides the pseudo-code of Algorithm 1 and  2, describing the workflow of FedIndex and DP mechanism.

---

**Algorithm 1** FedIndex

---

**Input:** initial encoder and predictor $\{E^{(0)}, F^{(0)}\}$, initial discriminator $D^{(0)}$; source domains $\{\mathcal{D}_i = \{x_i, y_i\}\}_{i \in [N]}$, target domain $\mathcal{D}_T = \{x_T\}$; source domain index set $\{\mathcal{U}_i\}_{i \in [N]}$, target domain index set $\{\mathcal{U}_T\}$; source clients $\{C_i\}_{i=1}^N$, target client $C_T$, server $S$; number of rounds $\mathcal{T}$

**Parameter:** lambda $\lambda_d$, Laplace parameter $b = \frac{\Delta f}{\epsilon}$

**Output:** global models $\{E_g, F_g\}$

1: $\{E_g^{(0)}, F_g^{(0)}\} = \{E^{(0)}, F^{(0)}\}$
2: $\tau_u \sim Lap(b)$
3: **for** $t = 1, 2, \ldots, \mathcal{T}$ **do**
4:     **for** $i = 1, 2, \ldots, N, T$ **do**
5:         $\{E_i^{(t-1)}, F_i^{(t-1)}\} = \{E_g^{(t-1)}, F_g^{(t-1)}\}$
6:         $z_i^{(t)} = E_i^{(t-1)}(x_i, u_i)$
7:         $\tilde{z}_i^{(t)} = z_i^{(t)} + \tau_i^{(t)}, \tau_i^{(t)} \sim Lap(b)$
8:         $\tilde{u}_i = u_i + \tau_u$
9:         $C_i$ uploads $\{\tilde{z}_i^{(t)}, \tilde{u}_i\}$ to server $S$
10:     **end for**
11:     Server S trains and obtains $D^{(t)}$ by the max function of Equation (1)
12:     **for** $i = 1, 2, \ldots, N, T$ **do**
13:         Server S sends $D^{(t)}$ back to client $C_i$
14:         **if** i = T **then**
15:             Client $C_i$ trains and obtains Encoder $E_i^{(t)}$ by the domain index loss in the min function of Equation (1)
16:             Client $C_i$ uploads the Encoder $E_i^{(t)}$ to the server
17:         **else**
18:             Client $C_i$ trains and obtains Encoder $E_i^{(t)}$ and Predictor $F_i^{(t)}$ by the min function of Equation (1)
19:             Client $C_i$ uploads the Encoder $E_i^{(t)}$ and Predictor $F_i^{(t)}$ to the server
20:         **end if**
21:     **end for**
22:     Server S aggregates the models by Equation (3) and broadcasts $\{E_g^{(t)}, F_g^{(t)}\}$
23: **end for**
24: **return** $\{E_g^{(\mathcal{T})}, F_g^{(\mathcal{T})}\}$

---

## B   Discussion

### B.1   Uploading Data to the Server in Federated Learning

*Why uploading data embeddings to the server in FL scenarios is acceptable?*

*First*, although FedIndex may introduce additional data transmission to enable continuous domain adaptation in FL by uploading data embeddings, existing studies on FDA (Peng et al., 2019b) and federated split learning (FSL) (Thapa et al., 2022) face similar challenges regarding extra data transmission. Thus, to enable FL for specific objectives, such as domain adaptation, it is common and acceptable to introduce extra data transmission.

*Second*, compared to FedAvg, FedIndex may require even less data transmission, since a FedIndex client need not upload the entire local model (*i.e.*, the local encoder, predictor, and discriminator). Assume

---

**Algorithm 2** DIfferential Privacy For FedIndex

---

**Input:** data $x$
**Parameter:** DP parameter $\epsilon$, sensitivity $\Delta f$
**Output**: noised data $\tilde{x}$

1: $b = \frac{\Delta f}{\epsilon}$
2: $\tilde{x} = x + \tau_x, \tau_x \sim Lap(b)$
3: **return** $\tilde{x}$

---

there are N clients, and each local dataset contains M samples with K-dimensional embeddings. For model components, the encoder has X parameters, the predictor has Y parameters, and the discriminator has Z parameters. Then, we can get the data transmission cost of FedIndex and FedAvg for uploading and downloading processes: 1) Uploading: FedIndex transmits N(X+Y+M×K) data to the server while FedAvg transmits N(X+Y+Z) data to the server. 2) Downloading: Both FedIndex and FedAvg download N(X+Y+Z) data from the server. Though a FedIndex client uploads M×K embeddings, it avoids uploading the local discriminator's Z parameters. Thus, whether FedIndex requires extra data transmission than typical FL depends on component and dataset settings. Please refer to Section 6 for the communication overhead of FedIndex on training datasets.

*Third*, FedIndex can also apply compression techniques, such as principal component analysis (PCA), to embeddings to reduce dimensionality and decrease data transmission while maintaining comparable prediction accuracy, since embeddings are sparse and contain many zeros.

### B.2 Incorporation of FedIndex and Orthogonal Safeguards

FedIndex raises additional privacy concerns compared to traditional FL methods, such as FedAvg, as it transmits data embeddings between clients and the server. In addition to model inversion attacks such as DIA and MIA, the transmission of data embeddings may also be vulnerable to model poisoning attacks, in which attackers transmit poisoned data embeddings to influence training and degrade overall performance. To defend against model poisoning attacks, FedIndex can incorporate safeguards such as Krum (Blanchard et al., 2017) to select the most typical data embeddings and then identify malicious poisoned data embeddings. Specifically, as the encoders extract the domain-invariant features, the distances between benign data embeddings are typically smaller than those between poisoned data embeddings. Hence, incorporating Krum could help FedIndex defend against model poisoning attacks that aim at degrading overall model performance (*e.g.*, classification accuracy).

### B.3 Training Stability of FedIndex with Non-IID Data

The training of FedIndex is not influenced by non-IID since the global discriminator can access privacy-preserving embeddings of data samples distributed across all clients. Taking one training round as an example: (1) Local encoders, initialized from the same global encoder of the previous round, generate and upload data embeddings to train the global discriminator. Even with non-IID data, the data embeddings are generated by encoders with the same weights. (2) The global discriminator is trained on embeddings from all clients' data samples, thus it is not influenced by the non-IID distribution, either. Furthermore, the previous study (Li et al., 2023) proves the convergence of the naive federated adaptation of adversarial learning (*i.e.*, employ a discriminator and an encoder on each client). Since FedIndex is not affected by non-IID data, it is more robust than prior work (Li et al., 2023) in terms of convergence guarantees under extreme skewness. Empirical results (Section 6) also demonstrates FedIndex's convergence under different non-IID settings.

### B.4 Impact of Federated Learning Training Dynamics on Theoretical Analysis

Although FedIndex adopts a similar analytical framework as prior studies (Zhao et al., 2019)—namely, analyzing component behavior at optimality, those studies are conducted under centralized settings, whereas

FedIndex operates in a FL environment. As such, our analysis must account for the unique training dynamics introduced by FL, particularly the effects of model aggregation.

To this end, we develop our analytical strategy through the following steps: (1) Identify the key property that FedIndex should exhibit at optimality, specifically, that synthetic data embeddings become indistinguishable from real data embeddings; (2) Design a learning framework capable of achieving this property, namely through an adversarial game between local encoders and discriminators; (3) Examine how FL training dynamics affect this framework. In particular, aggregating local discriminators across clients prevents the resulting global discriminator from achieving the same distinguishing capability as a centrally initialized discriminator, due to factors such as non-IID data and local update drift; (4) Adapt the learning framework to mitigate these effects by deploying a global discriminator on the server from the outset, rather than relying on locally trained discriminators.

Following this reasoning, we construct an adversarial learning framework comprising client-specific encoder–predictor pairs and a shared global discriminator that is robust to the aggregation-induced challenges inherent in FL.

## B.5 Deployments challenges

Previous work (Li et al., 2020a) has highlighted several deployment challenges faced by federated learning (FL) in real-world scenarios. In response, numerous studies (Li et al., 2018) have proposed methods to mitigate these challenges. In this work, we focus on the unique deployment challenges introduced by FedIndex, which are not typically encountered in conventional FL methods. Specifically, the additional data transmission required by FedIndex increases communication overhead, which may exacerbate resource constraints during real-world deployment.

To address this issue, we first empirically quantify the communication overhead of FedIndex in comparison to conventional FL methods such as FedAvg. Furthermore, we propose a practical approach to mitigate this overhead using data compression techniques, such as Principal Component Analysis (PCA). This is motivated by the observation that the latent-space representations used in FedIndex are typically sparse, with most elements close to zero.

Moreover, in real-world scenarios, domain indices may not always be available. To address this challenge, previous work (Xu et al., 2023) has proposed an efficient approach to generate domain indices.

## C    Theoretical Analysis

This section presents the corresponding theoretical analysis of DP and convergence of FedIndex.

### C.1    Theoretical Analysis of DP in Algorithm 1 and 2

A formal definition of Local DP is provided in Definition C.1. The privacy parameter $\epsilon$ captures the privacy loss incurred by the algorithm's output. Specifically, $\epsilon \to 0$ ensures perfect privacy in which the output is independent of its input, while $\epsilon \to \infty$ offers no privacy guarantee.

**Definition C.1** (Differential Privacy)**.** *Let $A : I \to O$ be a randomized algorithm mapping a data entry in $I$ to $O$. The algorithm $A$ is $(\epsilon, \delta)$-local differentially private if for all data entries $x, x' \in I$ and outputs $o \in O$ , we have:*

$$Pr\{A(x) = o\} \leq exp(\epsilon)Pr\{A(x') = o\} + \delta. \tag{8}$$

Algorithm 2 outlines how to add noise to inputs. The parameters $\Delta f$ and $\epsilon$ correspondingly are the dataset sensitivity and privacy budget. Measuring the true sensitivity of representation is challenging. Hence, we follow previous work bounding the sensitivity with 1 (*i.e.*, $\Delta f = 1$) (Shokri & Shmatikov, 2015). Theorem C.1 proposes a formal statement and proof of Algorithm 2's privacy guarantees.

**Theorem C.1** (Local Differential Privacy)**.** *Let the noise vector $\tau$ entries be drawn from $Lap(b)$ with $b = \frac{\Delta f}{\epsilon}$. Then Algorithm 2 is $\epsilon$-differentially private*

*Proof.* Let $f$ be a function that maps a dataset $I$ to some output. We denote the sensitivity $\Delta f$ to be:

$$\Delta f = max_{I,I'}||f(I) - f(I')||_1,$$

where $I'$ differs from $I$ at most one entry. Recall that the additive noise $\tau$ is drawn from Lap(b), $b = \frac{\Delta f}{\epsilon}$. Thus, the probability density function for $\tau$ is:

$$pdf(\tau) = \frac{1}{2b}exp(-\frac{|\tau|}{b}).$$

Given Definition C.1, we now expect Algorithm 1 (denoted as A) to be $\epsilon$-DP. In other words, for any subsets of possible outputs $S$, we have:

$$Pr\{A(I) \in S\} \leq e^\epsilon Pr\{A(I') \in S\},$$

where $A(I) = f(I) + \tau$ as given in Algorithm 1 and each $\tau_i$ is drawn from Lap(b). By denoting $A(I) = y, y \in S$ and replacing $\tau$, we can rewrite the probability density function of $Pr\{A(I) = y\}$ to be:

$$Pr\{A(I) = y\} = \prod_i \frac{1}{2b}exp(-\frac{|y_i - f(I)_i|}{b}).$$

Similarly, we could get the following equation:

$$Pr\{A(I') = y\} = \prod_i \frac{1}{2b}exp(-\frac{|y_i - f(I')_i|}{b}).$$

Then, we have:

$$\frac{Pr\{A(I) = y\}}{Pr\{A(I') = y\}} = \prod_i \frac{exp(-\frac{|y_i - f(I)_i|}{b})}{exp(-\frac{|y_i - f(I')_i|}{b})}$$
$$= \prod_i exp(\frac{|y_i - f(I')_i| - |y_i - f(I)_i|}{b}).$$

As defined, $I$ and $I'$ differ in at most one entry. Thus, we have:

$$\frac{|y_i - f(I')_i| - |y_i - f(I)_i|}{b} \leq \frac{|f(I)_i - f(I')_i|}{b}$$

and

$$\frac{|f(I)_i - f(I')_i|}{b} = \begin{cases} 0 & \text{if } f(I)_i = f(I')_i \\ \frac{\Delta f}{b} & o.w. \end{cases}$$

Only one term is non-zero as defined. Given $b = \frac{\Delta f}{\epsilon}$, we now have:

$$\frac{Pr\{A(I) \in S\}}{Pr\{A(I') \in S\}} = exp(\epsilon) \times \prod_{i-1} exp(0) \leq e^\epsilon.$$

$\square$

Now, we prove the local DP of each client. To demonstrate the global DP of Algorithm 1, Theorem C.2 provides a formal statement for Algorithm 1's privacy guarantees.

**Theorem C.2** (Global Differential Privacy). *Let clients employ Algorithm 2 during training. Then, Algorithm 1 is globally differentially private*

*Proof.* Given Algorithms 1 and 2, each client uploads and downloads data independently. Hence, global differential privacy is maintained during uplink/downlink procedures. The discriminator on the server processes the uploaded data individually, ensuring that differential privacy is not compromised. Therefore, the global differential of Algorithm 1 is achieved. $\square$

## C.2 Domain Alignment Ability Analysis of the Naive Solution

This subsection theoretically analyzes the naive solution (simply employing CIDA on each client in FL scenarios), and Theorem C.3 formally demonstrates that this naive approach yields a system with degraded domain-alignment ability compared to CIDA trained in centralized settings.

**Theorem C.3** (Sub-Optimal Performance of the Naive Solution). *Naively applying CIDA to each client in FL scenarios results in a system with diminished domain alignment. Due to non-convexity, local encoders align the conditional expectation of the domain index to the local marginal expectation within their respective domains, but fail to achieve global mean-alignment across all domains (i.e., $z \not\perp_1 u$).*

*Proof.* Simply employing CIDA on each client implies training one local discriminator $D_k$ per client $k$. As established in CIDA (Wang et al., 2020), each local discriminator $D_k$ minimizes the $L_2$ loss on its local dataset $\mathcal{D}_k$. Consequently, at equilibrium, the local encoder $E_k$ forces the conditional expectation to match the local marginal mean:

$$\mathbb{E}[u|z]_{z \in \mathcal{D}_k} = \mathbb{E}[u]_{\mathcal{D}_k}.$$

However, in a non-IID Federated Learning setting, the local marginal means differ across clients (*i.e.*, $\mathbb{E}[u]_{\mathcal{D}_k} \neq \mathbb{E}[u]_{\mathcal{D}_j}$ for $k \neq j$). The standard weight aggregation (FedAvg) of these local models results in a global encoder that represents a mixture of these disjoint alignments. Consequently, for the global data distribution, the conditional expectation $\mathbb{E}[u|z]$ varies depending on which client the data originated from, and does not converge to the single global marginal mean $\mathbb{E}[u]$. Thus, global mean-alignment is not achieved. $\square$

## C.3 Analysis of Simplified FedIndex

This subsection provides theoretical analysis for the simplified FedIndex, of which each local client consists of an encoder $E_i$ without a predictor $F_i$. The theoretical analysis follows the standard adversarial domain adaptation pattern, modeling a game in which the encoder aims to fool the discriminator, thereby preventing the discriminator from predicting the raw domain index.

In the absence of $F_i$, the optimization for $(E_i, D)$ is:

$$\max_{E_i} \min_D V_d(D, E_i) \tag{9}$$

$$= \mathbb{E}[L_d(D(E_i(x, u)) + \tau_i), u + \tau_u)], \tag{10}$$

where $\tau_i, \tau_u \sim Lap(b), b = \frac{\Delta f}{\epsilon}$.

Lemma C.1 below analyzes $D$ with the fixed $E$ and proposes that $D$ outputs the expectation of domain indices of all data with the same embedding $z$ added with noise $\tau_z$.

**Lemma C.1** (Optimal Discriminator for FedIndex). *For fixed $E$, the optimal $D$ is:*

$$D_E^*(E(x, u) + \tau_z) = \mathbb{E}_{u \sim p(u|\tilde{z})}[u],$$

*where $z = E(x, u)$, $\tilde{z} = z + \tau_i$, and $\tilde{u} = u + \tau_u$.*

*Proof.* With fixed $E$, the optimal $D$ is:

$$D_E^* = arg \min_D \mathbb{E}_{(x,u) \sim p(x,u), \tau_i, \tau_u}[L_d(D(E(x, u) + \tau_i), u + \tau_u)]$$

$$= arg \min_D \mathbb{E}_{(\tilde{z}, \tilde{u}) \sim p(\tilde{z}, \tilde{u})}[||D(z + \tau_i) - (u + \tau_u)||_2^2]$$

$$= arg \min_D \mathbb{E}_{\tilde{z} \sim p(\tilde{z})} \mathbb{E}_{\tilde{u} \sim p(\tilde{u}|\tilde{z})}[||D(z + \tau_i) - (u + \tau_u)||_2^2].$$

Given that

$$\mathbb{E}_{\tilde{u} \sim p(\tilde{u}|\tilde{z})}[||D(z + \tau_i) - (u + \tau_u)||_2^2]$$

$$= \mathbb{E}_{\tilde{u} \sim p(\tilde{u}|\tilde{z})}[(u + \tau_u)^2] - 2D(z + \tau_i)\mathbb{E}_{\tilde{u} \sim p(\tilde{u}|\tilde{z})}[u + \tau_u] + D(z + \tau_i)^2,$$

is a quadratic form of $D(z + \tau_i)$ which achieves optimum at

$$
\begin{aligned}
D(z + \tau_i) &= \mathbb{E}_{u \sim p(u|\tilde{z})}[u + \tau_u] \\
&= \mathbb{E}_{u \sim p(u|\tilde{z})}[u] + \mathbb{E}[\tau_u] = \mathbb{E}_{u \sim p(u|\tilde{z})}[u].
\end{aligned}
$$

$\square$

Assuming that $D$ always achieves the optimum w.r.t $E$, we can rewrite the optimization as:

$$
\max_{E_i} C_d(E_i),
$$

where

$$
\begin{aligned}
C_d(E_i) &\triangleq \min_D V_d(E_i, D) = V_d(E_i, D^*_{E_i}) \\
&= \mathbb{E}_{\tilde{z}} \mathbb{E}_{\tilde{u} \sim p(\tilde{u}|\tilde{z})} (\mathbb{E}_{\tilde{u} \sim p(\tilde{u}|\tilde{z})}[u + \tau_u] - (u + \tau_u))^2 \\
&= \mathbb{E}_{\tilde{z}} \mathbb{E}_{\tilde{u} \sim p(\tilde{u}|\tilde{z})}[u] + \mathbb{V}[\tau_u] = \mathbb{E}_{\tilde{z}} \mathbb{V}_{u \sim p(\tilde{u}|\tilde{z})}[u] + \mathbb{V}[\tau_u],
\end{aligned}
$$

where $\mathbb{V}$ denotes variance.

Then, we propose the theoretical analysis of the virtual training criterion $C_d(E_i)$ and the global optimum of simplified FedIndex.

**Lemma C.2** (Uniqueness of Constant Expectation). *If there exists a constant $\mu_c$ such that $\mathbb{E}_{u \sim p(u|\tilde{z})}[u] = \mu_c$ for any $\tilde{z}$, we have $\mu_c = \mathbb{E}_{u \sim p(u)}[u]$.*

**Theorem C.4** (Global Optimum for Simplified FedIndex). *With DP in FedIndex, $C_d(E_g)$ achieves the global optimum if and only if the encoder $E_g$ satisfies the expectations of the domain index $u$ over the conditional distribution $p(u|z)$ for any given $z$ are identical to the expectation over the marginal distribution $p(u)$, i.e., $\mathbb{E}[u|\tilde{z}] = \mathbb{E}[u], \forall \tilde{z}$.*

*Proof.* We first show $C_d(E_i) \leq \mathbb{V}[u] + \mathbb{V}[\tau_u]$ and then show the equality is achieved when $\mathbb{E}[u|z] = \mathbb{E}[u], \forall z$.

$$
\begin{aligned}
C_d(E_i) - \mathbb{V}[u] &= \mathbb{E}_{\tilde{z}} \mathbb{V}[u|\tilde{z}] - \mathbb{V}[u] + \mathbb{V}[\tau_u] \\
&= \mathbb{E}_{\tilde{z}}[\mathbb{E}[u^2|\tilde{z}] - \mathbb{E}[u|\tilde{z}]^2] - (\mathbb{E}[u^2] - \mathbb{E}[u]^2) + \mathbb{V}[\tau_u] \\
&= \mathbb{E}[u]^2 - \mathbb{E}_{\tilde{z}}[\mathbb{E}[u|\tilde{z}]^2] + \mathbb{V}[\tau_u].
\end{aligned}
$$

By the convexity of $x^2$ and Jensen's inequality, we have $\mathbb{E}[u]^2 = (\mathbb{E}_{\tilde{z}}[\mathbb{E}[u|\tilde{z}]])^2 \leq \mathbb{E}_{\tilde{z}}[\mathbb{E}[u|\tilde{z}]^2]$ and the equality is achieved when $\mathbb{E}[u|\tilde{z}]$ is constant w.r.t. z. By Lemma C.2, we have $\mathbb{E}[u|\tilde{z}] = \mathbb{E}[u], \forall \tilde{z}$. Then, given Equation 3, we have the same theorem for $C_d(E_g)$. $\square$

One key difference between FedIndex and CIDA lies in the additive Laplace noise introduced by FedIndex's DP on the data embedding and domain indices. Our Theorem C.4 proves that, despite the additive Laplace noise, the global encoder eventually aligns the domain indices from all domains.

**Corollary C.1.** *For FedIndex, the global optimum of $C_d(E_g)$ is achieved if the embeddings of all source domains are mean-aligned, i.e., $\tilde{z} \perp_1 u$.*

**Corollary C.2.** *For FedIndex, the Laplace noise introduced by DP does not alter the optimal alignment condition of embeddings of all domains, provided the noise has a mean of 0, i.e., $\mu(\tau) = 0$. The noise introduces a constant variance term $\mathbb{V}[\tau_u]$ to the objective function, which does not influence the gradients w.r.t. the encoder.*

## C.4 Analysis of FedIndex

This subsection provides FedIndex's equilibrium states in the three-player game of $E_i, F_i$, and $D$ as defined in Eqn. 1. We analyze the equilibrium behavior based on the dependency between the domain index $u$ and the label $y$.

$u \perp y$ (The Domain Index is a Nuisance Variable): The independence between the domain index $u$ and the label $y$ indicates that the domain index $u$ captures nuisance variance rather than useful information for prediction. Therefore, in this case, we prove the optimal global encoder captures all the information in the input $x$ relevant to the prediction while aligning the first moments (means) of the domain index distributions.

**Lemma C.3** (Optimal Predictor). *Given the encoder $E_i$, the prediction loss is lower-bounded by the conditional entropy:*

$$V_p(F_i, E_i) \stackrel{\triangle}{=} L_p(F_i(E_i(x, u)), y) \geq H(y|E_i(x, u)),$$

*where $H(\cdot)$ is the entropy. The optimal predictor $F_i^*$ that minimizes the prediction loss outputs the true posterior probability:*

$$F_i^*(E_i(x, u)) = P_y(\cdot|E_i(x, u)).$$

Assuming the predictor $F_i$ and the discriminator $D$ achieve their optimum by Lemma C.3 and Lemma C.1, the local objective function (Eqn. 1) can be rewritten as:

$$\min_{E_i} C(E_i) \stackrel{\triangle}{=} H(y|E_i(x, u)) - \lambda_d C_d(E_i). \tag{11}$$

**Theorem C.5.** *If the encoder $E_i$, the predictor $F_i$ and the discriminator $D$ are trained to reach optimum, any optimal local encoder $E_i^*$ and has the following properties:*

$$H(y|E_i^*(x, u)) = H(y|x, u) \tag{12}$$

$$C_d(E_i^*) = \max_{E_i'} C_d(E_i') \tag{13}$$

*Proof.* We establish a lower bound for the objective function $C(E_i)$. First, by the Data Processing Inequality, the entropy of the label given the embedding cannot be lower than the entropy given the raw data:

$$H(y|E_i(x, u)) \geq H(y|x, u).$$

Second, as derived in Appendix C.3, the alignment term $C_d(E_i)$ is bounded by the marginal variances:

$$C_d(E_i) \leq \max_{E'} C_d(E') = \mathbb{V}[u] + \mathbb{V}[\tau_u].$$

Combining these, the global minimum of the objective is bounded by:

$$C(E_i) \geq H(y|x, u) - \lambda_d(\mathbb{V}[u] + \mathbb{V}[\tau_u]).$$

The equality holds iff both conditions are met. We now argue that such an encoder exists. Let $E_0(x, u) = P(y|x, u)$. Then, (1) for prediction, since $E_0$ is the true posterior, it retains all mutual information with $y$, so $H(y|E_0) = H(y|x, u)$; (2) for alignment, since we assume $u \perp y$, the variable $y$ contains no information about $u$. Consequently, the optimal prediction features $E_0$ (which depend only on the relationship between inputs and $y$) are uncorrelated with $u$. Therefore, the conditional expectation of the domain index given the embedding is constant: $\mathbb{E}[u|E_0(x, u)] = \mathbb{E}[u]$. As established in Theorem C.4 and Corollary C.1, this condition implies $\mathbb{E}[u|\tilde{z}] = \mathbb{E}[u]$, which maximizes $C_d(E_i)$.

Thus, $E_i^*$ achieves the global optimum by simultaneously maximizing predictive accuracy and mean-alignment. $\square$

Theorem C.5 establishes that the optimal encoder retains all the information related to the label $y$ presented in the data $x$ and the domain index $u$ while simultaneously achieving alignment across different domains with noise data embedding and domain indices at equilibrium.

$u \not\perp y$ (The Domain Index is Informative): The domain index $u$ captures the information that helps predict label $y$. In this scenario, two terms in the objective function $C(E_i)$ conflict: (1) Minimizing $H(y|z)$ requires retaining information about $u$; (2) Maximizing $C_d(E_i)$ requires removing information about $u$. Consequently, FedIndex does not achieve perfect mean-alignment in this case. Instead, the hyperparameter $\lambda_d$ regulates a trade-off: the model learns a representation that is as domain-invariant as possible without sacrificing the predictive power derived from $u$. This prevents the model from relying on spurious domain correlations while retaining robust predictive features.

Table 7: Kolmogorov–Smirnov (KS) test results under different *normalized* privacy budgets $\epsilon$ (smaller $\epsilon$ indicates stronger privacy). Here, $\epsilon$ is reported under the normalized noise scale $b = 1/\epsilon$ for presentation consistency. The corresponding actual privacy budget should be scaled according to the actual dataset sensitivity, *i.e.*, $\epsilon_{\text{actual}} = \Delta f_{\text{actual}}\epsilon$. We report KS statistics (larger is better), and all corresponding $p$-values are less than 0.05.

| $\epsilon$ | Method | Half-Circle | Sine | RotatingMNIST | CompCar | BrainStroke | TCGA-BRCA | TPT-48 |
|---|---|---|---|---|---|---|---|---|
| $\epsilon = 1$ | Encoding | 0.32 | 0.30 | 0.29 | 0.38 | 0.35 | 0.36 | 0.40 |
| | Index | 0.28 | 0.26 | 0.25 | 0.33 | 0.31 | 0.33 | 0.35 |
| $\epsilon = 0.5$ | Encoding | 0.34 | 0.32 | 0.31 | 0.40 | 0.37 | 0.37 | 0.42 |
| | Index | 0.29 | 0.27 | 0.26 | 0.34 | 0.32 | 0.34 | 0.36 |
| $\epsilon = 0.3$ | Encoding | 0.36 | 0.34 | 0.33 | 0.42 | 0.39 | 0.39 | 0.44 |
| | Index | 0.30 | 0.28 | 0.27 | 0.35 | 0.33 | 0.32 | 0.37 |
| $\epsilon = 0.15$ | Encoding | 0.38 | 0.36 | 0.35 | 0.44 | 0.41 | 0.42 | 0.46 |
| | Index | 0.31 | 0.29 | 0.28 | 0.36 | 0.34 | 0.35 | 0.38 |
| $\epsilon = 0.1$ | Encoding | 0.40 | 0.38 | 0.36 | 0.45 | 0.42 | 0.43 | 0.48 |
| | Index | 0.32 | 0.30 | 0.29 | 0.37 | 0.35 | 0.36 | 0.39 |
| $\epsilon = 0.05$ | Encoding | 0.42 | 0.40 | 0.38 | 0.47 | 0.44 | 0.45 | 0.50 |
| | Index | 0.33 | 0.31 | 0.30 | 0.38 | 0.36 | 0.37 | 0.40 |
| $\epsilon = 0.01$ | Encoding | 0.45 | 0.43 | 0.41 | 0.50 | 0.47 | 0.47 | 0.53 |
| | Index | 0.35 | 0.33 | 0.32 | 0.40 | 0.38 | 0.39 | 0.42 |

## C.5 Convergence Analysis of FedIndex

Compared to conventional FL methods that employ adversarial training (*i.e.*, encoder versus discriminator), FedIndex preserves the same fundamental min-max training paradigm. The key modification is the privacy-preserving transmission of $\tilde{z}$ and $\tilde{u}$. Since the noise introduced by Differential Privacy has a constant variance and zero mean (as shown in Corollary C.2), it acts as a regularization term but does not alter the convexity/concavity properties of the underlying loss landscape with respect to the first moments. Therefore, the convergence analysis established in previous work for federated adversarial learning (Rasouli et al., 2020) remains applicable to FedIndex.

# D  Additional Experimental Results

This appendix provides the following experimental results:

- Visualization of Training Curves

- KS tests for DP effectiveness.

- Ablation study on DP and domain indices.

- Sensitivity study on $\lambda$, $\epsilon$, and *epoch*.

## D.1 Visualization of Toy Datasets

Figure 3 visualizes *Toy* datasets from the perspectives of all domains, all domains' label ground truth, target domains' label ground truth, and FedIndex's prediction.

## D.2 Effectiveness of DP

In this subsection, we examine the effectiveness of DP across all training datasets used in FedIndex. We define the dataset sensitivity $\Delta f$ under per-sample replace adjacency, *i.e.*, for any two inputs $x_i$ and $x_j$,

Table 8: Actual dataset sensitivity ($\Delta f_{\text{actual}}$) applied across different benchmarks.

| Sensitivity | Half-Circle | Sine | RotatingMNIST | CompCar | BrainStroke | TCGA-BRCA | TPT-48 |
|---|---|---|---|---|---|---|---|
| $\Delta f_{\text{actual}}$ | 13.34 | 12.78 | 17.45 | 29.76 | 22.41 | 17.64 | 7.5 |

Table 9: Sensitivity analysis of FedIndex with respect to the *normalized* privacy budget $\epsilon$, regularization weight $\lambda$, and training epochs. We report the mean $\pm$ standard deviation over multiple runs. Here, $\epsilon$ is reported under the normalized noise scale $b = 1/\epsilon$ for presentation consistency, while the corresponding actual privacy budget should be scaled by the empirical dataset sensitivity, *i.e.*, $\epsilon_{\text{actual}} = \Delta f_{\text{actual}} \epsilon$. Accuracy is reported for classification tasks, and MSE is reported for TPT-48.

| Setting | Value | Half-Circle Accuracy (%)↑ | Sine Accuracy (%)↑ | RotatingMNIST Accuracy (%)↑ | CompCar Accuracy (%)↑ | BrainStroke Accuracy (%)↑ | TCGA-BRCA Accuracy (%)↑ | TPT-48 MSE ↓ |
|---|---|---|---|---|---|---|---|---|
| $\epsilon$ | 0.15 | $86.5 \pm 6.84$ | $90.80 \pm 0.16$ | $82.56 \pm 3.24$ | $52.10 \pm 4.57$ | $63.5 \pm 3.91$ | $86.19 \pm 2.41$ | $0.43 \pm 0.03$ |
| | 0.30 | $87.2 \pm 7.70$ | $91.20 \pm 0.23$ | $84.09 \pm 2.81$ | $53.45 \pm 3.93$ | $63.09 \pm 3.70$ | $86.41 \pm 2.28$ | $0.42 \pm 0.02$ |
| | 0.50 | $87.6 \pm 7.46$ | $91.37 \pm 0.20$ | $85.35 \pm 3.01$ | $54.59 \pm 4.26$ | $64.40 \pm 4.19$ | $86.72 \pm 2.12$ | $0.42 \pm 0.02$ |
| | 1.00 | $87.80 \pm 7.11$ | $91.50 \pm 0.14$ | $87.40 \pm 2.48$ | $53.06 \pm 4.14$ | $65.57 \pm 3.61$ | $86.49 \pm 2.33$ | $0.40 \pm 0.02$ |
| $\lambda$ | 0.3 | $87.5 \pm 5.40$ | $93.85 \pm 0.14$ | $84.90 \pm 2.40$ | $55.64 \pm 4.72$ | $66.71 \pm 3.23$ | $85.78 \pm 1.66$ | $0.41 \pm 0.02$ |
| | 0.5 | $88.8 \pm 6.80$ | $90.88 \pm 0.23$ | $83.45 \pm 2.52$ | $53.49 \pm 3.84$ | $63.47 \pm 4.12$ | $88.92 \pm 2.84$ | $0.39 \pm 0.02$ |
| | 0.7 | $89.6 \pm 5.70$ | $92.74 \pm 0.13$ | $89.03 \pm 3.11$ | $54.70 \pm 4.31$ | $64.92 \pm 2.96$ | $85.18 \pm 1.92$ | $0.40 \pm 0.01$ |
| | 1.0 | $87.80 \pm 7.11$ | $91.50 \pm 0.14$ | $87.40 \pm 2.48$ | $53.06 \pm 4.14$ | $65.57 \pm 3.61$ | $86.49 \pm 2.33$ | $0.40 \pm 0.02$ |
| Epoch | 1 | $88.94 \pm 6.19$ | $91.50 \pm 0.14$ | $87.40 \pm 2.48$ | $53.06 \pm 4.14$ | $65.57 \pm 3.61$ | $86.49 \pm 2.33$ | $0.40 \pm 0.02$ |
| | 5 | $87.75 \pm 5.90$ | $91.0 \pm 0.30$ | $86.84 \pm 1.92$ | $53.5 \pm 3.74$ | $63.40 \pm 2.51$ | $86.2 \pm 1.83$ | $0.41 \pm 0.02$ |
| | 10 | $87.80 \pm 7.11$ | $91.7 \pm 0.20$ | $87.03 \pm 2.79$ | $55.0 \pm 1.71$ | $65.71 \pm 3.82$ | $86.9 \pm 1.97$ | $0.40 \pm 0.02$ |

$|x_i - x_j| \leq \Delta f$. We follow previous studies Dwork et al. (2006b) and set the dataset sensitivity to 1 when applying DP.

For domain indices, we normalize $u \in \mathbb{R}$ to $[0, 1]$ before training. Therefore, for any two domain indices $u_i$ and $u_j$, we have $|u_i - u_j| \leq 1$, which satisfies $\Delta f_{\text{actual}} \leq 1$.

For data embeddings, we instead empirically estimate the actual sensitivity, $\Delta f_{\text{actual}}$, on each training dataset. The results are reported in Table 8, which shows that the actual dataset sensitivities are not strictly bounded by 1. The maximum observed actual sensitivity across datasets is below 30. For consistent presentation at a fixed sensitivity, we report the privacy budget using a normalized parameter $\epsilon = \frac{\epsilon_{\text{actual}}}{\Delta f_{\text{actual}}}$. Under this notation, the actual noise scale $b = \frac{\Delta f_{\text{actual}}}{\epsilon_{\text{actual}}}$ can be equivalently written as $b = \frac{1}{\epsilon}$. Thus, the experiments reported here use the normalized form $b = 1/\epsilon$ only for the convenience of exposition, while corresponding to the actual privacy level determined by the empirical sensitivity bound.

To analyze the effectiveness of DP, we follow prior work (Wang et al., 2018; Awan & Wang, 2023) through Kolmogorov-Smirnov (KS) tests. Following this protocol, we apply KS tests with $\epsilon \in \{0.01, 0.05, 0.1, 0.15, 0.3, 0.5, 1\}$ in FedIndex to evaluate the effectiveness of DP. Table 7 reports the KS statistics and p-values for both data embeddings and domain indices under different privacy levels. In general, a small p-value ($\leq 0.05$) indicates that an attacker cannot confidently conclude that the privatized information and the raw transmitted information are drawn from the same distribution. This suggests that the transmitted information is effectively protected against DIA and MIA. Moreover, a smaller $\epsilon$ corresponds to stronger privacy protection, and all evaluated privacy budgets fall within a practically meaningful privacy regime.

### D.3 Sensitivity

For hyperparameter sensitivity:

- **Epsilon** ($\epsilon = [0.15, 0.3, 0.5, 1]$): Table 9 shows that the prediction accuracy of FedIndex decreases as the privacy level ($b = \frac{\Delta f}{\epsilon}$) increases.

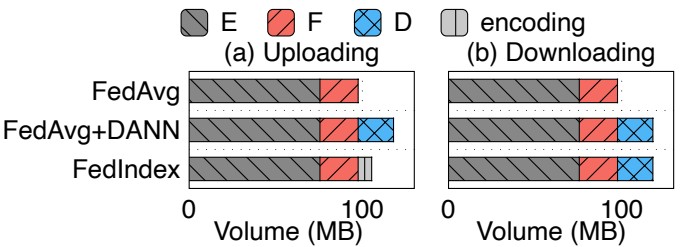

Figure 4: Average communication overhead (bytes) of FedIndex per round.

Table 10: Average communication overhead (seconds per round) under different scalability settings (number of clients). Lower is better.

| Model | #Clients | Half-Circle | Sine | RotatingMNIST | CompCar | BrainStroke | TCGA-BRCA | TPT-48 |
|---|---|---|---|---|---|---|---|---|
| FedIndex | 20 | 0.30 | 0.34 | 6.0 | 0.65 | 0.28 | 0.55 | 0.45 |
| | 40 | 0.35 | 0.39 | 12.0 | 0.85 | 0.34 | 0.75 | 0.60 |
| | 60 | 0.40 | 0.45 | 18.0 | 1.00 | 0.40 | 0.95 | 0.70 |
| | 80 | 0.48 | 0.53 | 25.0 | 1.20 | 0.52 | 1.15 | 0.90 |
| | 100 | 0.55 | 0.60 | 32.0 | 1.40 | 0.65 | 1.35 | 1.10 |
| FedAvg+DANN | 20 | 0.29 | 0.33 | 5.8 | 0.62 | 0.27 | 0.52 | 0.43 |
| | 40 | 0.34 | 0.38 | 11.5 | 0.82 | 0.33 | 0.72 | 0.58 |
| | 60 | 0.39 | 0.44 | 17.5 | 0.95 | 0.39 | 0.92 | 0.68 |
| | 80 | 0.47 | 0.51 | 24.0 | 1.15 | 0.50 | 1.10 | 0.88 |
| | 100 | 0.54 | 0.58 | 31.0 | 1.35 | 0.63 | 1.30 | 1.08 |
| FedAvg | 20 | 0.28 | 0.32 | 5.5 | 0.60 | 0.26 | 0.50 | 0.42 |
| | 40 | 0.33 | 0.36 | 11.0 | 0.80 | 0.32 | 0.70 | 0.56 |
| | 60 | 0.38 | 0.42 | 16.5 | 0.90 | 0.38 | 0.90 | 0.66 |
| | 80 | 0.46 | 0.49 | 23.0 | 1.10 | 0.48 | 1.05 | 0.86 |
| | 100 | 0.53 | 0.56 | 29.5 | 1.30 | 0.62 | 1.25 | 1.05 |

- **Epoch** ($epoch = [1, 5, 10]$): Table 9 shows that FedIndex requires more rounds to converge when *epoch* decreases.

- **Lambda** ($\lambda = [0.3, 0.5, 0.7, 1]$)**:** Table 9 shows that how $\lambda$ influences the prediction accuracy and MSE loss of FedIndex on training datasets. Empirically, $\lambda$ below one does not significantly impact the results.

## D.4 Communication Overhead

This section examines the average per-round communication overhead of FedIndex under varying scalability conditions, specifically evaluating configurations with 20, 40, 60, 80, and 100 clients. For comparison, we adopt the conventional FL method, FedAvg, as a baseline. *A detailed analysis of FedIndex's communication overhead, along with potential strategies for overhead reduction, is provided in Appendix B.1.*

Table 10 shows that FedIndex exhibits a comparable average communication overhead (s) per round to that of both FedAvg and FedAvg+DANN. Although FedIndex introduces additional transmission of data embeddings and continuous domain indices, it avoids uploading local discriminators by leveraging a centralized global discriminator initialized on the server. Furthermore, Table 10 demonstrates that the communication overhead of FedIndex scales approximately linearly with the number of participating clients. This linear trend can be attributed to hardware resource constraints, wherein certain clients must wait for available training slots, thereby inducing a proportional increase in communication time with client count. Furthermore, Figure 4 shows the communication overhead (byte) of FedIndex and baselines.

Table 11: Hyper-parameters and label-shift simulation configurations of FedIndex used in all training datasets. "General" represents the default value of a hyperparameter when no dataset-specific value is specified. For datasets with simulated label shift, we additionally report the corresponding generation strategy and its parameters, *e.g.*, the Dirichlet concentration parameter $\alpha$ or the label-flipping probability $p_{\text{flip}}$.

| Description | | General | Half-circle | Sine | RotatingMNIST | CompCar | BrainStroke | TCGA-BRCA | TPT-48 |
|---|---|---|---|---|---|---|---|---|---|
| *seed* | Value of the seed | 2333 | - | - | - | - | - | - | - |
| $\mathcal{T}$ | # of rounds | - | 60 | 150 | 50 | 200 | 300 | 200 | 500 |
| *epoch* | Local epochs | 1 | 10 | - | - | - | - | - | - |
| $B$ | Local batch size | 10 | - | 100 | 1000 | 30 | - | - | - |
| *lr* | Local learning rate | - | 1e-4 | 1e-5 | 2e-4 | 1e-4 | 5e-4 | 5e-4 | 1e-5 |
| $\lambda$ | Balance term | 1 | 0.3 | - | - | - | - | - | |
| $\Delta f$ | Dataset sensitivity | 1 | - | - | - | - | - | - | |
| $\epsilon$ | Privacy budget | 1 | - | - | - | - | - | - | |
| Label shift | Simulation strategy | - | Flipping | Flipping | Dirichlet | Dirichlet | Dirichlet | Dirichlet | Scaling |
| $\alpha$ | Dirichlet concentration | - | - | - | 0.5 | 0.8 | 0.5 | 0.5 | - |
| $p_{\text{flip}}$ / values | Flipping / shift parameters | - | [0.15, 0.21, 0.27, 0.33, 0.39, 0.45] | [0.25, 0.29, 0.33, 0.37] | - | - | | - | [0.3, 0.5, 0.7] |

# E    Hyperparameter Configurations

Table 11 summarizes the hyperparameter configurations of FedIndex across all datasets. In addition to the optimization-related hyperparameters, we also report the settings used to simulate label shift when applicable. Specifically, label shift is introduced either by sampling client-level class proportions from a Dirichlet distribution with concentration parameter $\alpha$, or by label flipping with flipping probability $p_{\text{flip}}$. Reporting these settings clarifies how label-distribution heterogeneity is constructed in each benchmark and improves the reproducibility of the experimental protocol.

# F    Reproduction

The source code and implementation details are available at `https://github.com/IntelliSys-Lab/FedIndex`.

