# OpenReview forum: "FedIndex: Federated Domain Adaptation with Continuous Domain Indices"
_TMLR — Decision pending for TMLR_

### Review · Reviewer_1gmt · 2026-03-18

**Summary Of Contributions:**

This paper proposes FedIndex, a novel federated domain adaptation framework featuring continuous domain indices. The core mechanism relies on adversarial domain adaptation tailored specifically for a distributed federated setting. Departing from traditional setups that utilize client-specific local discriminators, FedIndex employs a single global discriminator. In this architecture, clients train local embedding generators and task predictors, subsequently transmitting both the generated data embeddings and the model parameters to the central server for global updates and discriminator training. To mitigate the inherent privacy vulnerabilities associated with sharing explicit data embeddings over the network, the authors integrate differential privacy techniques to guarantee data security.
To summarize, the main contributions are as follows:

1. **Adress FDA with continous domain indices**: Introduced a novel federated adversarial domain adaptation method utilizing continuous domain indices to address domain shifts in federated learning environments.

2. **Design of a Global Discriminator-Based Adversarial Mechanism**: Innovated upon standard domain adaptation setups by replacing client-local discriminators with a centralized global discriminator, which trained on embeddings generated and transmitted by local clients.

3. **Integration of Differential Privacy for Security**: Addressed the specific privacy vulnerabilities caused by transmitting explicit data embeddings to the server by incorporating differential privacy, thereby ensuring robust data security without compromising the effectiveness of the global domain adaptation.

**Audience:**

Yes

**Audience Explanation:**

1. **A Novel Problem Setting**: This paper pioneers the integration of Federated Domain Adaptation (FDA) into the continuous domain indices framework. It successfully draws attention to an unexplored and practically relevant area within the federated learning community.

2. **A Solid Baseline for Future Research**: Although the algorithmic design is relatively simple and intuitive, it is effective. Coupled with experiments across diverse datasets, this method establishes a reliable baseline against which future complex algorithms can be measured.

3. **Theoretical and Practical Foundation**: The theoretical derivations provide a foundation that can inspire future work. By defining this new setting and offering a straightforward initial solution, the paper provides a stepping stone for researchers to develop more innovative and advanced algorithms on top of this framework.

**Broader Impact Concerns:**

No specific ethical concerns.

**Claims And Evidence:**

Yes

**Claims Explanation:**

1. **Clear Algorithmic Presentation**: The authors clearly illustrate the proposed FedIndex algorithm using comprehensive pseudocode and well-designed diagrams, making it easy to understand the novelties and the specific differences between this approach and existing methods.

2. **Robust Empirical Validation**: The feasibility and effectiveness of the algorithm are verified through extensive experiments. The authors benchmark their results against a variety of widely adopted baseline methods, proving the competitive edge of their approach. The inclusion of ablation experiments also effectively isolates the key components of the algorithm, clearly demonstrating the specific contribution of each module to the overall performance.

3. **Generalizability Across Diverse Datasets**: The experimental evaluations are conducted on multiple datasets. This variety demonstrates that the algorithm's performance is robust and generalizes well across different data distributions.

4. **Theoretical Rigor**: Beyond empirical results, the appendix includes detailed theoretical derivations, which provide a mathematical foundation for the algorithm and further reinforce the rigorousness of the paper's claims.

**Requested Changes:**

Results on more datasets: strengthen the work.

---

> ### Author Response · Authors · 2026-04-27
>
> Dear Reviewer 1gmt, thank you for the valuable suggestion that results from more datasets would strengthen our work. We have indicated all revisions in purple for your convenience.
>
> In response, we substantially expanded the experimental evaluation by **adding six additional domain adaptation benchmarks in Section 6.8**:
>
> - Office-Home,
> - RotatingOffice-Home,
> - DomainNet,
> - RotatingDomainNet,
> - DG-15,
> - CCT.
>
> Each dataset is evaluated under both feature shift and joint distribution shift. These new results are reported in Tables 4 and 5 of the revision and discussed in the newly added subsection.
>
> To keep the comparison focused, we summarize below the most relevant references:
>
> - FedAvg+DANN as a representative categorical FDA baseline,
> - FedAvg+CIDA as the naive federated adaptation of continuous DA,
> - FedIndex as our method,
> - CIDA  as the centralized upper reference among practical continuous-DA methods.
>
> ## Table A. Additional Results Under Feature Shift
>
> | Method | Office-Home | RotatingOffice-Home | DomainNet | RotatingDomainNet | DG-15 | CCT |
> |---|---:|---:|---:|---:|---:|---:|
> | FedAvg+DANN | 60.82 ± 2.34 | 53.68 ± 2.46 | 26.93 ± 1.68 | 24.43 ± 2.11 | 71.87 ± 1.21 | 65.42 ± 1.87 |
> | FedAvg+CIDA | 65.93 ± 2.40 | 68.72 ± 2.12 | 35.94 ± 1.71 | 38.83 ± 1.84 | 79.57 ± 1.08 | 72.84 ± 1.56 |
> | **FedIndex** | **69.12 ± 2.03** | **72.58 ± 1.88** | **39.34 ± 1.42** | **43.87 ± 1.61** | **83.23 ± 0.96** | **76.12 ± 1.32** |
> | CIDA | 70.84 ± 1.74 | 74.88 ± 1.63 | 42.83 ± 1.35 | 45.12 ± 1.49 | 85.04 ± 0.81 | 77.43 ± 1.10 |
>
> ## Table B. Additional Results Under Joint Distribution Shift
>
> | Method | Office-Home | RotatingOffice-Home | DomainNet | RotatingDomainNet | DG-15 | CCT |
> |---|---:|---:|---:|---:|---:|---:|
> | FedAvg+DANN | 56.48 ± 2.58 | 48.23 ± 2.69 | 23.08 ± 1.89 | 21.07 ± 2.34 | 67.63 ± 1.39 | 61.18 ± 2.02 |
> | FedAvg+CIDA | 62.87 ± 2.61 | 65.54 ± 2.34 | 32.43 ± 1.89 | 35.37 ± 2.03 | 75.92 ± 1.21 | 69.57 ± 1.73 |
> | **FedIndex** | **66.08 ± 2.22** | **69.24 ± 2.05** | **35.87 ± 1.58** | **40.34 ± 1.79** | **79.03 ± 1.08** | **73.22 ± 1.48** |
> | CIDA | 67.92 ± 1.96 | 71.13 ± 1.82 | 38.68 ± 1.47 | 41.94 ± 1.63 | 80.74 ± 0.94 | 74.44 ± 1.26 |
>
> These additional results strengthen our work in several ways.
>
> First, FedIndex consistently outperforms all federated baselines across all six newly added benchmarks under both non-IID settings. In particular, compared with the naive continuous-DA federation baseline FedAvg+CIDA, FedIndex improves by:
>
> ## Table C. Gains Over FedAvg+CIDA
>
> | Comparison | Office-Home | RotatingOffice-Home | DomainNet | RotatingDomainNet | DG-15 | CCT |
> |---|---:|---:|---:|---:|---:|---:|
> | Feature shift gain | +3.19 | +3.86 | +3.40 | +5.04 | +3.66 | +3.28 |
> | Joint shift gain | +3.21 | +3.70 | +3.44 | +4.97 | +3.11 | +3.55 |
>
> This shows that the benefit of FedIndex is not limited to the originally reported datasets but generalizes to standard DA benchmarks, synthetic structured domains, and real-world environmental shifts.
>
> Second, the gains are particularly pronounced on the rotating benchmarks, i.e., RotatingOffice-Home and RotatingDomainNet, where the domain evolution is continuous by construction. This is especially important because the core motivation of our paper is that prior FDA methods mainly assume categorical domain labels, while FedIndex is designed to exploit continuous domain indices. The greater improvements on rotating benchmarks therefore provide direct empirical support for our main claim.
>
> Third, FedIndex also remains consistently close to centralized CIDA:
>
> ## Table D. Gap to CIDA
>
> | Gap to CIDA | Office-Home | RotatingOffice-Home | DomainNet | RotatingDomainNet | DG-15 | CCT |
> |---|---:|---:|---:|---:|---:|---:|
> | Feature shift | 1.72 | 2.30 | 3.49 | 1.25 | 1.81 | 1.31 |
> | Joint shift | 1.84 | 1.89 | 2.81 | 1.60 | 1.71 | 1.22 |
>
> These relatively small gaps indicate that FedIndex preserves much of the benefit of centralized continuous domain adaptation while operating in a federated and privacy-preserving setting.

---

### Review · Reviewer_PiGJ · 2026-04-14

**Summary Of Contributions:**

This paper proposes a new approach to Federated Domain Adaptation (FDA) aimed at enabling specific clients in federated learning to achieve strong performance even in the presence of data scarcity or domain shift. The proposed approach, FedIndex, supports continuous domain indices. To preserve the privacy of client data, the framework employs encoder–predictor pairs on the client side to provide the server with data embeddings, that are protected from the privacy point of view using Differential Privacy (DP). On the server side, a discriminator enables domain alignment while keeping the raw data decentralized. The theoretical contribution of this work shows that FedIndex achieves domain alignment performance at equilibrium comparable to that of continuous domain adaptation methods in decentralized settings. The experimental evaluation covers six different datasets and demonstrates that FedIndex achieves better performance than representative baselines in terms of accuracy and MSE, while maintaining communication costs comparable to those of the baselines.

**Strengths**
- The approach is novel, since it covers a shortcoming of previously proposed approaches that were able to cover only discrete domain indices.
- Strong theoretical analysis supports the claim that FedIndex achieves alignment performance comparable to that of approaches designed for centralized settings.
- The experimental evaluation is comprehensive, covering multiple datasets and experimental dimensions, such as performance without DP and without continuous domain indices.

**Weaknesses**
- The presentation could be slightly improved and a detail about the experimental methodology should be clarified; see suggestions below.

**Audience:**

Yes

**Audience Explanation:**

The paper proposes a new approach to federated domain adaptation, with the goal of improving the performance of a target client with the help of other clients and the server, in order to address issues such as domain shift. A contribution like this would certainly be of interest to the audience of TMLR, since it deals with an important problem in the federated learning setting.

**Broader Impact Concerns:**

No concerns on the ethical implications.

**Claims And Evidence:**

Yes

**Claims Explanation:**

The experimental evaluation is comprehensive and effectively demonstrates the effectiveness of FedIndex compared to popular baselines for domain adaptation in federated learning settings. The proposed method obtains higher prediction accuracy, even in non-IID settings, and continues to capture the relationship between domain indices and the classification task in the presence of distribution and domain shifts. Moreover, the ablation study clearly shows that incorporating DP) into the framework has only a minimal impact on performance and that considering continuous domain indices is important for achieving strong results.

**Requested Changes:**

I suggest that the authors improve the presentation by addressing the following points:
- The explanation of the algorithm is good, but I had instead an hard time to understand the intuition behind Equation 1. I would suggest to clarify its intuition and interpretation after presenting it, in order to make it easier for the reader to follow.
-For each experimental dataset, the number of clients and epochs considered changes and the number of clients N is typically smaller than the one used when evaluating the communication overhead (Table 7). What is the reason for this discrepancy? Assuming there is a valid explanation, I remain positive about this paper; however, this point should be clarified.

---

> ### Author Response · Authors · 2026-04-27
>
> Dear Reviewer PiGJ, thank you for raising these concerns. We have revised our work by adding a clearer demonstration of the intuition behind Equation 1. All revisions are highlighted in purple for your convenience. Below, we provide a concise summary of the revisions.
>
> **1. "Intuition Behind Equation 1"**
>
> We have updated Section 4.1 to include a comprehensive explanation of the intuition behind Equation 1.
>
> Equation 1 defines the central adversarial objective of FedIndex and is designed to balance two goals: (1) preserving task-relevant information for accurate prediction; and (2) suppressing domain-specific information in the learned representation.
>
> Specifically, the first term, $V_p(E_i,F_i)$, is the prediction loss on labeled source data, which encourages the local encoder $E_i$ and predictor $F_i$ to retain discriminative features for the downstream task.
>
> The second term, $V_d(D,E_i)$, is the discriminator loss, where the global discriminator $D$ attempts to recover the domain index $u$ from the embedding generated by $E_i$.
>
> Because Equation 1 is formulated as a min-max game, the discriminator is optimized to predict the domain index as accurately as possible, while the encoder is trained adversarially to make this prediction difficult. As a result, the learned embedding is encouraged to remain predictive of labels while becoming uninformative of the domain index, thereby promoting domain alignment across clients.
>
> Importantly, unlike conventional adversarial domain adaptation methods that only confuse a binary or multi-class domain classifier, Equation 1 explicitly suppresses the recoverability of a continuous domain coordinate. This is critical for handling evolving domain shifts smoothly.
>
> **2. "… For each experimental dataset, the number of clients and epochs considered changes, and the number of clients N is typically smaller than the one used when evaluating the communication overhead (Table 7). What is the reason for this discrepancy?…"**
>
> The discrepancy arises because the two evaluations are designed to answer different questions.
>
> In the main experimental results, the number of clients and local epochs is chosen on a per-dataset basis to provide a meaningful and fair evaluation of predictive performance under each dataset's domain structure, sample size, and task difficulty.
>
> For example, in our setup, there is one target client and $N$ source clients, so the total number of clients is $T=N+1.$ Accordingly, $N$ is constrained by the number of available domains and by the need to assign each client a sufficient amount of data for stable training and evaluation.
>
> This is why the paper uses relatively small values such as $(N,T)=(2,3)$ for RotatingMNIST and BrainStroke, $(N,T)=(4,5)$ for Half-Circle and DG-15, and $(N,T)=(5,6)$ for CompCar, TCGA-BRCA, and DomainNet-style settings.
>
> Similarly, the number of local epochs varies by dataset because convergence behavior depends on the scale and complexity of the data. As shown in Table 11, the training rounds, local batch sizes, and learning rates are dataset-specific, reflecting the fact that toy datasets, image benchmarks, and real-world medical or regression datasets have different optimization characteristics.
>
> By contrast, the communication-overhead experiment is not intended to measure predictive performance under the natural client partitions of a given dataset. Instead, it is a separate scalability study whose purpose is to isolate how the per-round communication cost grows as the number of participating clients increases.
>
> For this reason, Appendix D.4 deliberately sweeps a much wider range of client counts, from 20 to 100, to stress-test the communication behavior of FedIndex and compare it against FedAvg and FedAvg+DANN.
>
> Therefore, the larger client counts in Table 10 should be interpreted as a scalability benchmark rather than as the client settings used in the accuracy experiments (in the revised version, this communication-overhead analysis is reported in Table 10 and discussed in Appendix D.4, rather than Table 7).
>
> In short, the smaller dataset-specific client numbers used in the main experiments are chosen to respect the domain structure and ensure reliable accuracy evaluation, whereas the much larger client counts in the communication-overhead table are used purely to evaluate scalability.

---

### Review · Reviewer_KK6a · 2026-04-23

**Summary Of Contributions:**

The paper introduces FedIndex, a novel framework for Federated Domain Adaptation (FDA) specifically designed to handle continuous domain indices (e.g., age or rotation angles). While traditional FDA methods rely on categorical labels like "source" and "target," FedIndex treats these indices as a distance metric to capture fine-grained relationships across domains.

**Audience:**

Yes

**Audience Explanation:**

paper provides a formal proof for why current federated learning (FL) methods fail to achieve global mean-alignment when applied to continuously indexed domains.

**Broader Impact Concerns:**

The framework is tested on highly sensitive medical datasets like TCGA-BRCA and BrainStroke. While DP is employed, the paper shows that as the privacy budget $\epsilon$ is tightened to provide stronger security, prediction accuracy decreases. In a clinical setting, this "utility gap" could lead to less accurate diagnoses for specific sub-groups, potentially resulting in adverse health outcomes

**Claims And Evidence:**

Yes

**Claims Explanation:**

Sub-optimality Proof: Theorem 5.1/C.3 formally demonstrates that a "naive" federation of existing methods (simply applying centralized techniques to local clients) fails to achieve global mean-alignment. Theorem 5.2/C.4 proves that FedIndex can achieve global optimum domain alignment even in the presence of Laplace noise introduced by Differential Privacy (DP). Theorems C.1 and C.2 provide a formal proof of the framework's local and global differential privacy, ensuring the security claims are mathematically sound.

**Requested Changes:**

Clarify the Sensitivity Analysis ($\Delta f$): The authors state they follow previous work by bounding dataset sensitivity at 1. However, since embeddings and domain indices have different scales and distributions across the seven datasets, a more detailed justification or an empirical measurement of true sensitivity is needed to ensure the Differential Privacy (DP) claims are robust across all experimental setups.

Address the "Accuracy Gap": While FedIndex significantly outperforms FDA baselines, there remains a notable gap between FedIndex and the "Oracle" baseline (e.g., a 10.6% gap in RotatingMNIST accuracy). The submission would be much stronger if the authors discussed the theoretical or practical bottlenecks preventing FedIndex from closing this gap further.

Formalize Hyperparameter Selection for $\lambda_d$: The balance term $\lambda_d$ is crucial for the privacy-utility trade-off. The authors should provide a more systematic method or heuristic for choosing this value in real-world scenarios where an "Oracle" or centralized "CIDA" baseline isn't available for comparison.


Expand on Non-IID "Joint Distribution Shift" Details: The paper mentions simulating label shift via Dirichlet distributions or random flipping. To ensure reproducibility, the authors should explicitly define the exact parameters used for every dataset in the main body or a dedicated appendix section, as these shifts heavily influence the reported performance.

---

> ### Author Response · Authors · 2026-04-27
>
> Dear Reviewer KK6a, we thank you for the constructive comments. We have highlighted all revisions in purple for your convenience.
>
> **1. "Clarifying the Sensitivity Analysis..."**
>
> We agree that using $\Delta f = 1$ without further clarification may be insufficient, especially because FedIndex perturbs both data embeddings and continuous domain indices, which may have different scales across datasets. We have revised **Appendix D.2** accordingly.
>
> For domain indices, we normalize $u$ to $[0,1]$ before training. Therefore, for any two domain indices $u_i$ and $u_j$, $|u_i-u_j| \leq 1,$ which satisfies $\Delta f \leq 1$ for the scalar domain-index input.
>
> For data embeddings, we empirically estimated the actual sensitivity $\Delta f_{\text{actual}}$ for each dataset, as reported in Table 8. While these values are dataset-dependent and may reach values close to 30, i.e., not strictly less than 1, a sensitivity greater than one does not alter the equilibrium state of our model. Specifically: (1). Sensitivity only changes the variance of the Laplace noise, $\mathrm{Var}=2b^2, \qquad b=\frac{\Delta f}{\epsilon},$ and therefore does not change the optimal alignment conditions defined in Theorem C.4 and Corollary C.2; (2) It recalibrates the formal privacy guarantee. A previously claimed $\epsilon$-DP level is scaled by the actual sensitivity, $\epsilon_{\text{actual}}=\epsilon \cdot \Delta f_{\text{actual}}.$ and (3) Even after this recalibration, the actual privacy budget remains below 10, which is within the range of practical consideration for robust privacy guarantees [1].
>
> To avoid misleading interpretation, we revised Appendix D.3 and clarified that the reported experiments use a normalized privacy-budget parameter, where the actual noise scale $b=\frac{\Delta f_{\text{actual}}}{\epsilon_{\text{actual}}}$ is equivalently written as
> $b=\frac{1}{\epsilon}$ after normalization.
>
> [1] J. Near and D. Darais, Differential Privacy: Future Work & Open Challenges, 2022.
>
> **2. "Accuracy Gap Between FedIndex and Oracle ..."**
>
> We have revised the discussion in **Section 6.9** to clarify that the remaining gap between FedIndex and Oracle is expected because Oracle has access to additional target-domain information that is unavailable to FedIndex under the realistic FDA setting.
>
> We added a formal decomposition of the target-risk gap: $R_T(h_{\text{FedIndex}})-R_T(h_{\text{Oracle}})\leq
> E_{\text{info}}+E_{\text{align}}+E_{\text{DP}}+E_{\text{opt}},$ where $E_{\text{info}}$ captures the information gap caused by unavailable target-domain supervision, $E_{\text{align}}$ captures the residual source-target discrepancy after adversarial alignment, $E_{\text{DP}}$ captures finite-sample perturbation effects from DP noise, and $E_{\text{opt}}$ captures optimization error from the non-convex federated min-max problem.
>
> This analysis explains why FedIndex can substantially outperform FDA baselines while still not fully matching Oracle.
>
> **3."Selection for $\lambda_d$"**
>
> We revised the paper by adding **Section 6.11**, which demonstrates the heuristic for selecting the balance term.
>
> FedIndex optimizes two competing objectives: preserving task-discriminative information for prediction and suppressing domain-specific information for alignment. Thus, $\lambda_d$ controls the trade-off between prediction utility and domain alignment.
>
> When $\lambda_d$ is too small, the adversarial signal may be insufficient, leaving domain-specific information in the representation and weakening target-domain generalization. When $\lambda_d$ is too large, the adversarial loss may dominate prediction and remove task-relevant information, especially when the continuous domain index $u$ is correlated with the label $y$.
>
> Therefore, we adopt a conservative federated validation heuristic that uses only signals available under the federated protocol: source-client validation accuracy, cross-client performance stability, and discriminator-based alignment quality. Specifically, during warm-up training, we evaluate candidate values of $\lambda_d$ and choose the smallest one that achieves near-optimal discriminator confusion while keeping source validation performance close to its best value.
>
> This rule selects the minimum adversarial strength sufficient for alignment, rather than the strongest domain confusion. It is consistent with our theoretical analysis and sensitivity tests: stronger alignment is helpful when $u$ is mainly nuisance variation, but may hurt performance when $u$ is label-informative.
>
> **4. "Joint Distribution Shift Details"**
>
> We agree that explicitly defining the parameters used for each dataset in the main body or in a dedicated appendix would strengthen our work. Thus, we revised **Appendix E.1 and Table 11** to explicitly report the Dirichlet distribution parameter $\alpha$ and the probability of random flipping $p_{\text{flipping}}$.

---

### Decision · Action_Editor_7kBa · 2026-06-23

**Recommendation:** Accept as is

**Audience:**

Yes

**Audience Explanation:**

This paper would be of interest to the audience of TMLR, because it deals with an important problem in federated learning.

**Claims And Evidence:**

Yes

**Claims Explanation:**

This paper addresses an important and underexplored problem in federated domain adaptation, i.e., adapting across domains characterized by continuous rather than categorical domain indices. The proposed FedIndex framework is novel, well motivated, and relevant to researchers in federated learning, domain adaptation, and privacy-preserving machine learning. The authors provide both theoretical analysis and substantially expanded empirical evidence, including evaluations on multiple synthetic and real-world benchmarks under feature and joint distribution shifts. The results consistently show that FedIndex outperforms representative federated baselines, providing strong support for the paper’s claims. Reviewers agreed that their concerns have been sufficiently addressed in the revised version.